# Beyond Laplace and Gaussian: Exploring the Generalized Gaussian Mechanism for Private Machine Learning

## Abstract

Differential privacy (DP) is obtained by randomizing a data analysis algorithm, which necessarily introduces a tradeoff between its utility and privacy. Many DP mechanisms are built upon one of two underlying tools: Laplace and Gaussian additive noise mechanisms. We expand the search space of algorithms by investigating the Generalized Gaussian (GG) mechanism, which samples the additive noise term $x$ with probability proportional to $e^{-\frac{|x|^{\beta}}{\sigma}}$ for some $\beta \geq 1$. The Laplace and Gaussian mechanisms are special cases of GG for $\beta = 1$ and $\beta = 2$ respectively.

In this work, we prove that all members of the GG family satisfy differential privacy, and provide an extension to an existing numerical accountant (the PRV accountant) to do privacy accounting. We apply the GG mechanism to two canonical tools for private machine learning, PATE and DP-SGD; we show that $\beta$ has a weak relationship with test-accuracy, and that $\beta = 2$ (Gaussian) is often a near-optimal value of $\beta$ for the privacy-accuracy tradeoff of both algorithms. This provides justification for the widespread adoption of the Gaussian mechanism in DP learning. That said, we do observe a minor improvement in the utility of both algorithms for $\beta \neq 2$, suggesting that further exploration of general families of noise distributions may be a worthy pursuit to improve performance in DP mechanisms.

## 1 Introduction

As applications of machine learning (ML) often involve sensitive information, there is an increasing need to provide privacy protections for the individuals whose data are included in the training datasets. Privacy concerns have prompted the development of privacy-preserving ML techniques, which aim to prevent the leakage of private information analyzed during training. One of the primary frameworks for achieving this goal is differential privacy (DP), a mathematical framework that provides quantifiable privacy guarantees (Dwork et al., 2006).

Two popular techniques for implementing DP in ML are Differentially Private Stochastic Gradient Descent (DP-SGD) (Abadi et al., 2016) and Private Aggregation of Teacher Ensembles (PATE) (Papernot et al., 2017). Traditionally, DP-SGD entails Poisson sampling from the dataset, gradient clipping, and then the addition of Gaussian noise to the gradient. PATE, on the other hand, involves training an ensemble of teacher models on disjoint subsets of the data, then privately aggregating the votes of the teacher models on a public dataset, and training a student model on the privately labeled public dataset. PATE achieves private vote aggregation through a private variant of Argmax, which is obtained by adding noise to the vote counts, and then finding the Argmax of the noisy histogram.

Both DP-SGD and PATE achieve privacy protection through mechanisms that add noise drawn from specific probability distributions, e.g., Laplace or Gaussian. The choice of this distribution plays a crucial role in determining the privacy-accuracy tradeoffs of an algorithm, and algorithm designers often make problem-dependent decisions in choosing between Laplace and Gaussian Mechanisms. However, many of the underlying tradeoffs between these two discrete choices remain unclear. In this work, we explore a continuum of private mechanisms that extends these two special cases of noise distributions. We investigate the Generalized Gaussian Mechanism (GG) (Liu, 2019), denoted $GG_{\beta,\sigma}(f, D)$, which adds noise to the true function value $f(D)$ sampled from the Generalized

Gaussian distribution[1], denoted $\mathcal{N}_\beta(\mu, \sigma)$, with probability density function (PDF),

$$p(x|\mu, \sigma, \beta) \propto e^{-\frac{|x-\mu|^\beta}{\sigma}}. \tag{1}$$

We focus on the GG Mechanism because it generalizes both the Laplace and Gaussian Mechanisms, which are are special cases of the GG Mechanism for $\beta = 1$ and $\beta = 2$, respectively.

In Figure 1 we show several PDFs of the $\mathcal{N}_\beta(0, \sigma)$ for different $\beta$ and $\sigma$ values (and $\mu = 0$) corresponding to GG Mechanisms that satisfy equivalent $(\epsilon, \delta)$ DP guarantees. For more details on this, we explore this further in Section 3.2 and Appendix B.4.

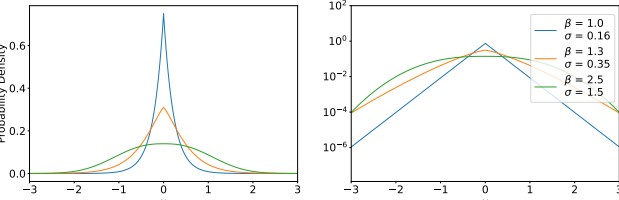

Figure 1: Linear (left) and log-scale (right) PDFs of the Generalized Gaussian distribution corresponding with GG mechanisms that satisfy an equivalent $(\epsilon, \delta)$-DP guarantee. For an equivalent value of $\sigma$, a larger value of $\beta$ yields a PDF that is more concentrated around the mean; yet, in order to satisfy the same $(\epsilon, \delta)$-DP, one must simultaneously increase $\sigma$ to compensate for a lighter tail.

We explore two settings relevant to ML: PATE and DP-SGD. Our findings show that the the choice of $\beta$ has a relatively small effect of test-accuracy, but that values around $\beta = 2$ perform near-optimal. This helps explain why Gaussian noise may be so popular, but suggests that the choice of Gaussian or Laplace Mechanism plays a smaller role than anticipated. Yet, for settings where small improvements are critical, hyperparameter searching of $\beta$ can provide important, incremental gains. Importantly, we find the Gaussian mechanism does not appear to be completely optimal, and thus our work also suggests future directions for further hyperparameter search in DP algorithm design.

## 1.1 RELATED WORK

There exist numerous alternative DP mechanisms. The Staircase Mechanism (Geng & Viswanath, 2013) and Podium Mechanism (Pihur, 2019) were derived as alternatives to the Laplace Mechanism by minimizing the variance of the noise distribution in order to improve utility guarantees. While both mechanisms have rigorous analysis of utility in the single-shot regime, they are neither studied nor optimized for high-composition regimes, and are not used in ML for this reason.

Alghamdi et al. (2022) developed the Cactus Mechanism to address the high-composition regime by numerically computing a mechanism that minimizes the Kullback-Leibler divergence between the conditional output distributions of a mechanism given two different inputs, under a high number of compositions. However, they only considered privacy for 1-dimensional outputs, making the result not applicable to ML models. Further, their mechanism is only optimal in the high-composition setting. Awan & Dong (2022) addresses the multivariate setting by deriving a family of multivariate log-concave canonical noise distributions (Awan & Vadhan, 2023), allowing addition of minimal noise for a particular f-DP guarantee. However, their work is constrained to $\ell_1$ and $\ell_\infty$ sensitivity settings and does not analyze the privacy-utility tradeoff of these mechanisms for private ML. Unfortunately, the proposed mechanisms are either only for Gaussian DP (known to underestimate privacy (Gopi et al., 2021)), or are not easily integrated with existing privacy accountants in the subsampled regime.

Liu (2019) introduced and partially analyzed the Generalized Gaussian Mechanism, which adds noise from the Generalized Gaussian distribution (Definition 7). They provided probabilistic-DP guarantees for $\beta \geq 2$, and gave empirical results for integer $\beta$-values in the GG mechanism applied to training Support Vector Machines on tabular datasets. Separately, Liu et al. (2021), provides a relatively weak bound for the Rényi Differential Privacy (RDP 6) when $\beta > 2$, for the limited use case of $\alpha = 1$, which restricts much of the usefulness of the RDP formulation. To best of our knowledge, no previous work provides tight $(\epsilon, \delta)$-DP guarantees for the GG Mechanism that are useful in the high composition regime. In our work, we consider all $\beta \geq 1$, and explore the privacy-accuracy tradeoff

---

[1]Sometimes referred to as the Exponential Power Distribution or Generalized Normal Distribution.

of the GG mechanism and its applications to PATE and DP-SGD, and empirically provide tighter privacy guarantees than those provided by prior works, for any number of compositions.

## 1.2 OUR CONTRIBUTIONS

In this work, we investigate the Generalized Gaussian Mechanism, a little-explored family of mechanisms that satisfies DP. We also introduce the Sampled Generalized Gaussian Mechanism (SGG), which is a variant of the GG-Mechanism that involves first subsampling the database to make use of privacy amplification by subsampling. In Section 4, we introduce the GGNMax algorithm for computing a private argmax. And in Section 5, we show how to use the Sampled GG Mechanism for DP-SGD in a new mechanism that we name $\beta$-Differentially Private Stochastic Gradient Descent ($\beta$-DP-SGD). We show that all 4 mechanisms satisfy differential privacy (Theorems 3.1, 4, 4.1, and A.2). We also show how to extend an existing privacy accountant (the PRV accountant) to track privacy budget over many compositions of these mechanisms.

In Section 4, we empirically find that in privately computing an argmax in PATE, the choice of $\beta$ has a weak relationship with the accuracy. In Section 5, we find that $\beta$-DP-SGD performs similarly for most $\beta \in [1, 4]$ when hyperparameters can be tuned based on the choice of $\beta$. Yet, small improvements do exist, typically around the neighborhood of $\beta \approx 2$, which can be valuable when accuracy is very critical; this effect is more noticeable when specific hyperparameters are fixed.

One key finding of our work is that although the Gaussian mechanism ($\beta = 2$) achieves near-optimal performance in these application domains, it is not exactly optimal in all regimes, thus suggesting space for further optimization. Previously, explorations of new mechanisms were confined to those noise distributions that exhibited well-behaved mathematical properties (such as analytically derivable Rényi Divergence values between distributions); this work develops a framework for using the PRV accountant (Gopi et al., 2021) to explore new mechanisms, which can enable new directions for research in novel DP mechanisms that cannot be directly analyzed analytically.

## 2 PRELIMINARIES

### 2.1 DIFFERENTIAL PRIVACY

Differential privacy (DP) is a framework for designing privacy-preserving data analysis algorithms that protect the privacy of individuals in a dataset while allowing accurate statistical analysis. Informally, DP provides a mathematical guarantee that an individual's data will have only a limited affect on the result of analysis on a large database. Two datasets are said to be *neighboring* if they differ only in a single data record.

**Definition 1** (Differential privacy (Dwork et al., 2006)). *A mechanism $\mathcal{M} : \mathcal{D} \rightarrow \mathcal{R}$ satisfies $(\epsilon, \delta)$-differential privacy if for any two neighboring datasets $D, D' \in \mathcal{D}$ and for any $S \subseteq \mathcal{R}$,*

$$\Pr[\mathcal{M}(D) \in S] \leq e^\epsilon \cdot \Pr[\mathcal{M}(D') \in S] + \delta.$$

Smaller values of the parameters $\epsilon$ and $\delta$ correspond to stronger privacy guarantees. The Laplace and Gaussian Mechanisms are examples of *output perturbation* mechanisms, which first evaluate a function on the input dataset, and then add mean-zero noise to the result. The variance of the noise scales with the *sensitivity* of the function $\Delta f$, defined as the maximum change in the function's value due to the removal or addition of a single database entry: $\Delta f = \max_{D, D' \text{ neighbors}} |f(D) - f(D')|$.

Two of the most common mechanisms in DP are the Laplace Mechanism and the Gaussian Mechanism.

**Definition 2.** *The Laplace Distribution (centered at 0) with scale b is the distribution with probability density function:*

$$Lap(x|b) = \frac{1}{2b} \exp(-\frac{|x|}{b}).$$

**Definition 3** (Laplace Mechanism (Dwork et al., 2006)). *For any $\epsilon > 0$, given a real-valued function $f : \mathcal{D} \rightarrow \mathbb{R}$, the Laplace mechanism is defined as*

$$\mathcal{M}_L(D, f, \epsilon) = f(D) + Y,$$

*where $Y \sim Lap(\Delta f / \epsilon)$. The Laplace Mechanism is $(\epsilon, 0)$-DP.*

**Definition 4** (Gaussian Mechanism (Dwork & Roth, 2014)). *For any $\epsilon > 0$ and $\delta \in (0, 1]$, given a real-valued function $f : \mathcal{D} \to \mathbb{R}$, the Gaussian mechanism is defined as*

$$\mathcal{M}_G(D, f, \epsilon) = f(D) + Y,$$

*where $Y \sim \mathcal{N}(0, \sigma)$ for $\sigma > \Delta f \sqrt{2\log(1.25/\delta)}/\epsilon$. The Gaussian Mechanism is $(\epsilon, \delta)$-DP.*

One main feature of differential privacy is that the guarantees *compose*, meaning that the overall privacy loss (as measured by $\epsilon$ and $\delta$) of running multiple DP mechanisms can be bounded as a function of the privacy parameters of the individual mechanisms. In the simplest version of composition (Dwork et al., 2006), the privacy parameters "add up" so that running two $(\epsilon, \delta)$-DP mechanisms results in $(2\epsilon, 2\delta)$-DP overall. In practice, however, this naive composition dramatically overestimates the incurred privacy risk, and more advanced composition algorithms (Dwork et al., 2010) and privacy accountants (Abadi et al., 2016) are used to more accurately bound privacy risk.

A privacy accountant is a tool to track the privacy budget of a system by recording the privacy cost associated with each query; accountants are particularly important for applications like DP-SGD, where DP mechanisms are composed a large number of times, e.g., as many as the number of steps in gradient descent. The introduction of the Moments Accountant (Abadi et al., 2016) enabled the first use of DP-SGD with reasonable privacy guarantees on common datasets like MNIST (Lecun et al., 1998). This was later replaced in many settings by accountants that rely on Renyi Differential Privacy (RDP), introduced by Mironov (2017).

## 2.2 RÉNYI DIFFERENTIAL PRIVACY

Rényi Differential Privacy (RDP) generalizes pure differential privacy ($\delta = 0$) and is closely related to the moments accountant. Defined below, the RDP guarantee of a mechanism is stated in terms of Rényi divergence.

**Definition 5** (Rényi Divergence). *The Rényi divergence of order $\alpha$ between two distributions $P$ and $Q$ is defined as:*

$$D_\alpha(P||Q) = \frac{1}{\alpha - 1}\log \mathbb{E}_{x \sim Q}\left[(P(x)/Q(x))^\alpha\right] = \frac{1}{\alpha - 1}\log \mathbb{E}_{x \sim P}\left[(P(x)/Q(x))^{\alpha - 1}\right].$$

**Definition 6.** *(Rényi Differential Privacy (Mironov, 2017)). A randomized mechanism $\mathcal{M}$ satisfies $(\alpha, \epsilon)$-RDP with $\alpha \geq 1$ if for any neighboring datasets $D$ and $D'$:*

$$D_\alpha(\mathcal{M}(D)||\mathcal{M}(D')) = \frac{1}{\alpha - 1}\log \mathbb{E}_{x \sim \mathcal{M}(D)}\left[\left(\frac{\Pr[\mathcal{M}(D) = x]}{\Pr[\mathcal{M}(D') = x]}\right)^{\alpha - 1}\right] \leq \epsilon.$$

RDP is desirable in ML applications because of its straightforward composition properties: the adaptive composition of mechanisms $\mathcal{M}_1, \ldots, \mathcal{M}_k$ where each $\mathcal{M}_i$ satisfies $(\alpha, \epsilon_i)$-RDP, will together satisfy $(\alpha, \sum_{i=1}^k \epsilon_i)$-RDP.

Pure $(\epsilon, 0)$-DP corresponds to $(\infty, \epsilon)$-RDP; Mironov (2017) provided more general guarantees for converting between RDP and DP: if $\mathcal{M}$ is an $(\alpha, \epsilon)$-RDP mechanism, it also satisfies $(\epsilon + \frac{\log(1/\delta)}{\alpha - 1}, \delta)$-DP for any $\delta \in (0, 1)$.

## 2.3 PRV ACCOUNTANT

The privacy guarantees of a DP mechanism can be defined as a function $\delta(\epsilon)$, since the probability of failure ($\delta$) generally depends on the required $\epsilon$-bound. This naturally leads to the definition of a privacy curve, such that for every $\epsilon \in \mathbb{R}$, $\mathcal{M}$ is $(\epsilon, \delta(\epsilon))$-DP for the appropriate function $\delta(\epsilon)$. Gopi et al. (2021) provided an efficient method for composing privacy curves directly that gave much tighter privacy guarantees, using an accountant called the Privacy Random Variable accountant (PRV). This relies on a connection between a DP mechanism's privacy curve $\delta(\epsilon)$ and its uniquely defined privacy loss random variables $(X, Y)$, which represent the likelihood of returning a particular outcome on two neighboring databases, respectively defined as:

$$X = \log(\frac{Q(\omega)}{P(\omega)}) \text{ where } \omega \sim P; \quad Y = \log(\frac{Q(\omega)}{P(\omega)}) \text{ where } \omega \sim Q,$$

where $P$ and $Q$ are the distribution of the mechanism's output over two neighboring datasets. Intuitively, the privacy loss random variables can be thought of as the *actual* $\epsilon$ value for a specific output; it is a random variable because the output $\mathcal{M}(D)$ is itself a random function.

Gopi et al. (2021) introduced the algorithm ComposePRV, which efficiently computes the privacy guarantees for the composition of multiple DP mechanisms. ComposePRV takes as input the CDFs of PRVs $Y_1, \ldots, Y_k$ (as well as a few other hyperparameters), and returns an estimate of the privacy curve for all the mechanisms composed, represented by $\delta(\epsilon)$, allowing for the direct computation of $\epsilon$.

## 3 GENERALIZED GAUSSIAN MECHANISM AND PRIVACY GUARANTEES

We first introduce the Generalized Gaussian (GG) Mechanism and show that it satisfies DP (Section 3.1). Since these privacy results are existential, rather than descriptive – i.e., we show that *there exists* some $\epsilon$ and $\delta$ values, rather than providing a closed form relationship between $(\beta, \sigma)$ and $(\epsilon, \delta)$ – we also present a PRV-based privacy accounting method (Section 3.2) that can be used to measure explicit $(\epsilon, \delta)$-DP guarantees when applying this mechanism to ML tasks, (see Sections 4 and 5).

### 3.1 GENERALIZED GAUSSIAN MECHANISM

We first formally define the Generalized Gaussian distribution and introduce the Generalized Gaussian Mechanism (Algorithm 1), which is an output perturbation mechanism that adds noise sampled from the Generalized Gaussian distribution.

**Definition 7** ((Dytso et al., 2018)). *The Generalized Gaussian distribution, denoted $\mathcal{N}_\beta(\mu, \sigma)$, is specified by the pdf $p(x|\mu, \sigma, \beta) \propto e^{-\frac{|x-\mu|^\beta}{\sigma}}$ with normalizing constant $\frac{\beta}{2\sigma^{\frac{1}{\beta}}\Gamma(\frac{1}{\beta})}$.*

---

**Algorithm 1** Generalized Gaussian Mechanism, $GG_{\beta,\sigma}(f, D)$. (Ganesh & Zhao, 2020)

---

1: **Input:** noise parameters $\beta \geq 1$, $\sigma > 0$, vector-valued function $f : \mathcal{D} \to \mathbb{R}^d$, database $D \in \mathcal{D}$.
   Let $\Delta_2 f = \max_{D, D' \text{ neighbors}} \|f(D) - f(D')\|_2$
2: **for** $i = 1$ to $d$ **do**
3:     Sample $Y_i \sim \mathcal{N}_\beta(0, \sigma \cdot \Delta_2 f)$
4: **end for**
5: **Output:** $f(D) + (Y_1, \ldots Y_d)$

---

Next, Theorem 3.1 states that the GG Mechanism satisfies DP. Critically, this theorem is only a claim about existence of a DP bound; this is because the guarantee that the mechanism is DP for *some* $\epsilon$ and $\delta$ is sufficient to apply the PRV accountant described in Section 3.2.

**Theorem 1.** *For any dataset $D \subset \mathbb{R}^{n \times d}$, and any function $f : \mathbb{R}^{n \times d} \to \mathbb{R}$, with sensitivity $\Delta f \geq 0$, then $\forall \beta \geq 1, \forall \sigma > 0$, the $GG_{\beta,\sigma}(f, D)$ is differentially private.*

We leave the proof of Theorem 3.1 in Appendix C.2.

In Appendix A, we introduce and analyze the Sampled Generalized Gaussian Mechanism, SGG (Algorithm 2), which is a variant of the GG Mechanism that first applies Poisson subsampling to the input database, evaluates the function $f$ on the sample, and then adds Generalized Gaussian noise to the result. This mechanism is motivated by *privacy amplification by subsampling*, which is popular in ML applications and strengthens privacy guarantees without increasing the level of noise added by the mechanism, by subsampling the database before applying a DP mechanism.

### 3.2 PRIVACY ACCOUNTING FOR GG MECHANISMS

In this work we focus on the PRV accountant because: it empirically provides tighter guarantees than other accountants (Gopi et al., 2021), it is implemented in common codebases such as Opacus (Yousefpour et al., 2021), and we are able to extend the PRV accountant to work for privacy accounting of arbitrary DP mechanisms such as the GG mechanism, which do not typically exist in closed-form.

In order to calculate the privacy consumed using a PRV accountant, one must solely input the CDF of the PRV, along with a few hyperparameters. In Appendix C.4, we extend the known PRVs for Laplace and Gaussian Mechanisms and compute a closed-form expression for the PRVs of the Generalized Gaussian Mechanism, which enables us to apply the PRV accountant.

We compute the privacy guarantees of $GG_{\beta,\sigma}(f, D)$ using the PRV accountant by sampling from the appropriate $\mathcal{N}_\beta(\mu, \sigma)$ distribution and numerically computing the CDF of $Y$. (Gopi et al., 2021) provides a tight estimate of the error in the PRV accountant's estimate; our work extends this computation to provide a similarly tight error analysis bounding the contribution of error from sampling to the estimate, which is included inAppendix B.3. We pass the sampled CDF as input to the ComposePRV algorithm of Gopi et al. (2021), which takes the CDFs of the PRVs of the composed mechanisms, and returns a composed privacy curve $\delta(\epsilon)$, providing an $\epsilon$ value for a specified choice of $\delta$. More implementation details of this change to the PRV accountant are in Appendix C.4.

Figure 2 illustrates the resulting value of $\epsilon$ as a function of $\sigma$ for different values of $\beta$ and fixed values of $\delta = 10^{-5}$ and $\Delta f = 1$. All curves have a similar shape as known privacy curves for Gaussian ($\beta = 2$) and Laplace ($\beta = 1$) Mechanisms. Additionally, as $\beta$ grows, the same $(\epsilon, \delta)$-DP guarantee necessitates a larger value of $\sigma$. Here, we show the privacy curves for a single composition of the GG mechanism, but importantly, these privacy curves and their relative differences change with the number of times the mechanisms are composed.

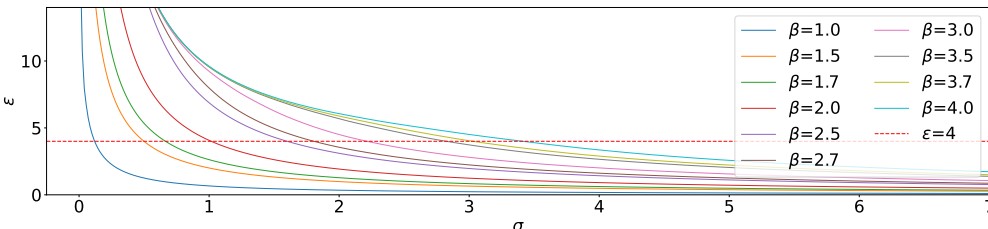

Figure 2: DP parameter $\epsilon$ as a function of noise parameter $\sigma$ with fixed $\delta = 10^{-5}$ and $\Delta f = 1$, calculated using the PRV accountant. Mechanisms with equivalent DP guarantees can be identified by computing a privacy curve's intersection with a horizontal line, illustrated here with a red line for (arbitrarily chosen) $\epsilon = 4$.

Remarkably, when using sensitivity defined in the $\ell_\beta$ norm we observe that the privacy cost of the multi-dimensional mechanism is equivalent to the single dimension mechanism. This makes privacy accounting for $GG_{\beta,\sigma}(f, D)$ dimension independent with regards to some $\ell_\beta$ senstivity of $f$. We prove this in result Appendix B.2, and also provide an analytic solution for the one dimensional GG mechanism in Appendix C.3.

Further, for a fixed privacy budget $(\epsilon, \delta)$, varying $\beta$ will subsequently change the weight of the tails of the distributions. Thus, for a 1-dimensional definition of "outlier", it is possible to derive the optimal GG mechanism that minimizes the likelihood of outliers; we explore this in Appendix B.5.

## 4 GG MECHANISM FOR PATE AND PRIVATE ARGMAX

Private Aggregation of Teacher Ensembles (PATE) (Papernot et al., 2017) is an algorithm for training a private machine learning model. In PATE, a dataset is partitioned and a model is trained on each partition, then the models in the ensemble privately vote on the labels of an unlabeled dataset, and finally a model is trained on the privately labeled dataset; see Appendix D.2 for a more detailed explanation. The core step in PATE that provides formal privacy is the private computation of the Argmax over the votes of an ensemble of models. This is done based on a variation of the ReportNoisyMax algorithm, and is the center of our focus in this section.

PATE has been extensively studied with variations of Laplace and Gaussian Mechanisms; Papernot et al. (2017) employed LNMax, which privately aggregates votes from an ensemble of models by taking the argmax of a histogram after adding Laplace noise. Later, Papernot et al. (2018) developed several variations based on adding Gaussian noise, including GNMax, and empirically found them to be superior to their Laplace counterpart. We introduce a new algorithm, GGNMax, generalizing the GNMax and LNMax algorithms, which adds noise from the Generalized Gaussian Distribution $\mathcal{N}_\beta(0, \sigma)$. We show that the effect of $\beta$ on average label accuracy is relatively weak and Laplace and

Gaussian noise work produce nearly equivalent privacy-accuracy tradeoffs. We supplement these findings with simulations in Appendix D.3, and we empirically show that the Gaussian mechanism is near-optimal when the correct label of the histogram aligns with the majority vote.

## 4.1 PRIVATE ARGMAX AND THE GGNMAX MECHANISM

We present our Generalized Gaussian Private Argmax algorithm (GGNMax), which takes in noise parameters $(\beta, \sigma)$, a set of real-valued functions with sensitivity $\Delta$, and a database. The algorithm adds noise sampled from $\mathcal{N}_\beta(0, \sigma\Delta)$ to each coordinate and then returns the index of the coordinate with the largest value. We provide a definition below and include an algorithm in Algorithm 5.

**Definition 8.** *Generalized Gaussian Private Argmax(GGNMax) GGNMax$(\beta, \sigma, \{f\}, \Delta, D)$ is defined as the $\arg\max_{i \in [N]}\{f_i(D) + Y_i\}$, where $Y_i \sim \mathcal{N}_\beta(0, \sigma\Delta)$; for noise parameters $\beta \geq 1$, $\sigma > 0$, functions $f_1, \ldots, f_N : \mathcal{D} \to \mathbb{R}$ each of sensitivity $\Delta$, database $D \in \mathcal{D}$.*

Importantly, like the Laplace-based Report Noisy Max algorithm (Dwork & Roth, 2014) and the Gaussian-based GNMax algorithm (Papernot et al., 2018), the privacy of the GGNMax mechanism satisfies $(\epsilon, \delta)$-DP guarantees of the GG mechanism in 1 dimension. We state this formally in Theorem 2 and prove this in Appendix C.5.

**Theorem 2.** *If the $(\beta, \sigma)$-Generalized Gaussian Mechanism is $(\epsilon, \delta)$-DP for a fixed $\epsilon > 0$ and $\delta \geq 0$, then $(\beta, \sigma)$-Generalized Gaussian Private Argmax is also $(\epsilon, \delta)$-DP.*

## 4.2 PATE EXPERIMENTS

In order to study the GGNMax algorithm's application to PATE, we evaluate the effect of $\beta$ on the label accuracy of the GGNMax algorithm when applied to the histograms produced in an intermediate step of PATE. To study this for a realistic setting, we use the histograms generated on the MNIST and the Street View House Numbers (SVHN) dataset (Netzer et al., 2011), produced by the Papernot et al. (2017), which introduced the PATE algorithm. For each of these datasets, we start with a collection of $10,000$ histograms; each histogram is the collection of 250 models trained on a partition of the dataset, and evaluated on an unlabeled datapoint $x$. Then, for each histogram, we compute the private label produced by the GGNMax mechanism for 20 evenly spaced values $\beta \in [1, 4]$ and 100 values of $\sigma \in [0.01, 7]$. For each fixed $(\epsilon, \delta)$ and value of $\beta$, we compute the average label accuracy with respect to the ground truth labels provided by the dataset, averaged across 25 trials.

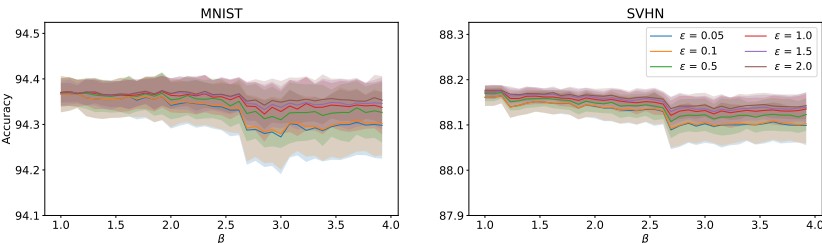

Figure 3: Average label accuracy of GGNMax mechanisms with equivalent privacy guarantees and varying values of $\beta$, evaluated on histograms which were generated by Papernot et al. (2017) as an intermediate state produced by 250 teachers trained on MNIST (left) and SVHN (right)

Figure 3 shows the average label accuracy for the GGNMax mechanism applied to histograms generated as part of PATE for MNIST and SVHN, as a function of $\beta$. We observe that values of $\beta$ in the region of $\beta \in [1, 2.5]$ perform roughly equivalently, and for larger choices of $\beta$, the label accuracy decreases slightly. [2]

In Appendix D.3, we investigate the privacy-accuracy tradeoff of the GGNMax algorithm on simulated histograms where the true label is the same as the argmax of the unnoised histograms. In this simulated setting, we find that $\beta$ values close to 2 perform near-optimally.

---

[2]We observe a small drop in performance around $\beta = 2.6$, which we believe is an artifact of how the mechanisms of equivalent DP guarantees are generated. First, we add noise from an evenly distributed grid of $\beta \in [1, 4]$ and $\sigma \in [0.01, 7]$ values, and then we compute the privacy for each $(\beta, \sigma)$ tuple; only then do we compute the corresponding mechanisms of approximately equal DP guarantees. This can cause us to add more noise than is required, particularly for larger $\beta$.

## 5 GG MECHANISM FOR DP-SGD

We now turn to Differentially Private Stochastic Gradient Descent (DP-SGD), which is one of the most commonly used mechanisms for private ML. We propose a simple change to DP-SGD: replacing the Gaussian noise used in DP-SGD with Generalized Gaussian noise and using the $\ell_\beta$ norm for clipping rather than $\ell_2$[3]. We call the resulting mechanism the $\beta$-Generalized Gaussian Differentially Private SGD ($\beta$-DP-SGD). This is corresponds to changing the underlying mechanism in DP-SGD from the Sampled Gaussian Mechanism to the Sampled Generalized Gaussian Mechanism (SGG), as DP-SGD performs Poisson subsampling on the dataset before computing the gradient update and adding Gaussian noise. This can be presented as replacing the gradient update step in the original DP-SGD algorithm with $\tilde{G}_t \leftarrow \frac{1}{L}\left(\sum_i \bar{G}_t(x_i) + \vec{Y}\right)$, where $\vec{Y}$ is computed as a vector as follows $Y_1, \ldots, Y_d \sim_{i.i.d.} \mathcal{N}_\beta(0, \sigma \cdot C)$. We present $\beta$-DP-SGD formally in Algorithm 6. We prove that $\beta$-DP-SGD is DP and its privacy can be computed using an accountant for the SGG Mechanism with appropriate hyperparameters in Appendix A.2

### 5.1 DP-SGD EXPERIMENTS

We seek to find a relationship between $\beta$ and DP-SGD's privacy-accuracy trade-off for non-convex optimization tasks by comparing test-accuracy as a function of $\epsilon$, for different $\beta$. It is only possible to study this relationship empirically, because even in the absence of privacy, theoretical bounds on the performance of SGD are not known for general non-convex optimization tasks. Further, we use the PRV accountant because it outperforms accountants that provide closed-form privacy guarantees (like the RDP accountant), but it only provides a numerically computed privacy guarantee.

To provide a robust evaluation of the role of $\beta$ in the $\beta$-DP-SGD algorithm, we focus on 4 datasets in different domains: CIFAR-10 (Krizhevsky, 2009) and Street View House Numbers (SVHN) (Netzer et al., 2011), two common computer vision datasets; the Adult dataset (Becker & Kohavi, 1996), a tabular dataset with a binary classification task; and the IMDB dataset (Maas et al., 2011), a collection of movie reviews meant for binary sentiment classification. We train four different architectures; for the vision classification tasks, we use the models described in in Tramèr & Boneh (2020), which previously achieved SOTA results for the $\epsilon \leq \sim 2.5$ regime (ScatterNet CNNs)[4]. For the the Adult Dataset we train a 2-layer Fully Connected Network (FCN). And for the IMDB dataset, we train a Long-Short Term Memory (LSTM) network with $\sim$ 1M parameters.

A full description of the hyperparameters, datasets, and models is included in Appendix E.2. Each experiment is run 3 times, which we found sufficient given standard deviations that generally fell below 0.3%. We find that when fixing a choice of $\beta$ and allowing for hyperparameter tuning along all other hyperparameters, we see a weak but noticeable relationship with final test accuracy.

**Results:** To isolate the effect $\beta$ in $\beta$-DP-SGD we report the maximum test-accuracy achieved by each architecture, by extracting the maximum across all hyperparameters, and we report the standard deviation across 3 trials for each set of hyperparameters. Our $\beta$-DP-SGD algorithm produces a relatively weak relationship between $\beta$ and test accuracy, but it is more noticeable in lower $\epsilon$ regimes (high privacy). We present this in Figure 4.

Figure 4 presents the the maximum test accuracy for 3 different values of $\epsilon$, evaluated for 3 different models on 4 different datasets (ScatterNet on CIFAR-10 and SVHN, a FCN on the Adult dataset, and an LSTM on the IMDB dataset), across different values of $\beta$. Similar to our results with the GGNMax, we find that for all $\epsilon$ values tested, the choice of $\beta$ has a weak relationship with final test accuracy, across most values of $\epsilon$. However, unlike the results in Section 4, $\beta$-DP-SGD seems to perform worse for larger values of $\beta$, particularly for values larger than $\beta \geq 3.0$. In Appendix E.3, we explore the relationship of individual hyperparameters with $\beta$ and find a weak, but more noticeable effect of $\beta$ on the final test accuracy, particularly for larger $\epsilon$ [5].

---

[3]We choose to use $\ell_\beta$ clipping rather than a fixed choice like $\ell_2$ because when using the $GG_{\beta,\sigma}(f, D)$ mechanism with $\ell_\beta$ sensitivity, privacy accounting dimension-independent, as proven in Appendix B.2.

[4]We also train an ordinary CNN for the vision tasks, but leave these results in Appendix E.3

[5]We leave a description of this in the appendix, as full-hyperparameter search is common.

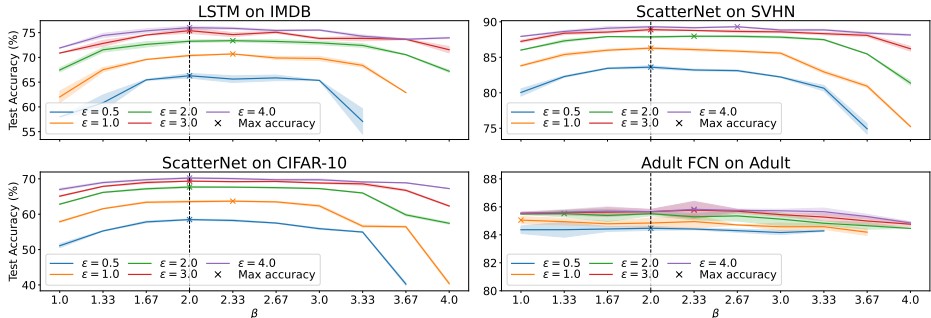

Figure 4: $\beta$-DP-SGD results for different architectures trained on CIFAR-10, SVHN, Adult, and IMDB, for $\delta = 10^{-6}$. The test-accuracy is reported for 3 values of $\epsilon$, computed for each architecture and dataset. A vertical dashed line denotes the Gaussian mechanism. Note: Some values are not presented (for lower $\epsilon$), because larger $\beta$ tends to consume more privacy per step, and the model's privacy budget exceed the target in the first step.

Importantly, despite a relatively weak relationship between $\beta$ and test-accuracy, $\beta$-DP-SGD is able to reach – and slightly surpass – SOTA results, in the example case of CIFAR-10, presented in Table 1. Before directly comparing results, we observe that most existing SOTA results use privacy guarantees provided by an RDP accountant, which can overestimate privacy loss relative to PRV accounting. In order to disambiguate empirical differences due to improved accounting versus the GG mechanism, we present our recreated SOTA results with the PRV accountant, alongside the results for $\beta$-DP-SGD.

| Architecture | Accountant | Training Algorithm | $\epsilon = 1$ | $\epsilon = 2$ | $\epsilon = 3$ | $\beta$ **value** |
|---|---|---|---|---|---|---|
| sCNN | RDP | DP-SGD | 60.3 (-) | 67.2 (-) | 69.3 (-) | - |
| WRN-40-4 | RDP | DP-SGD | 56.4 (0.6) | 65.9 (0.5) | **70.7** (0.2) | - |
| sCNN | PRV | DP-SGD | 63.5 (0.4) | 67.6 (0.1) | 69.2 (0.3) | - |
| sCNN | PRV | $\beta$-DP-SGD (*ours*) | **63.7** (0.1) | **67.7** (0.1) | 69.4 (0.1) | (2.33, 2.33, 2) |

Table 1: SOTA Results for private ML, evaluated on CIFAR-10, grouped by privacy accountant. The $\beta$ value column is set to '-' if trained with traditional DP-SGD, otherwise it reports a tuple of $\beta$ values that achieve max accuracy for $\epsilon = 1, 2, 3$, respectively (ties broken by smaller std values). sCNN is a CNN model ($\sim$ 5e5 parameters) trained on Scatternet features, and WRN-40-4 is a Wide Resnet ($\sim$ 1e7 parameters).

While Figure 4 shows that the choice of $\beta$ in $\beta$-DP-SGD has relatively minimal effect on the final test accuracy for a model, Table 1 reveals that for models that produce SOTA (or nearly-SOTA) test accuracies, the minor improvements in performance that result from optimizing $\beta$ may push the test accuracy beyond the existing SOTA. Though this needs to be traded off against the privacy cost of hyperparameter search (Papernot & Steinke, 2022).

## 6 CONCLUSION

We studied the Generalized Gaussian Mechanism, its privacy guarantees, and its applications to private ML, particularly for PATE (via private Argmax) and DP-SGD. This work reveals that the choice of $\beta$ has a relatively modest influence on test accuracy, and the difference between Gaussian, Laplace, and other GG Mechanisms is smaller than anticipated. Interestingly, values close to $\beta = 2$ exhibit near-optimal performance, which provides insight into the popularity of Gaussian noise in DP-SGD, PATE, and other private ML applications. Our observations that the Gaussian is not always exactly optimal in specific settings suggests new opportunities for the design of DP mechanisms.

An interesting extension for future work is that our GG Mechanism – as well as its variants SGG, GGNmax, and $\beta$-DP-SGD – sample noise independently across dimensions. For the Laplace Mechanism ($\beta = 1$), it is known that sampling from a high-dimensional Laplace variant can improve performance in private ML settings such as private empirical risk minimization (Chaudhuri et al., 2011). Interestingly, multi-dimensional Gaussian distributions are the only spherically symmetric distribution, where all the component random variables are independent Ali (1980), so such high-dimensional variants would not improve performance for $\beta = 2$. This suggests that for $\beta \in [1, 2)$, it may be possible to significantly improve utility for the same privacy guarantee by sampling from a single high-dimensional distribution rather than sampling independently for each coordinate.

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

## A  Sampled Generalized Gaussian Mechanism

*Privacy amplification by subsampling* is a technique to strengthen DP guarantees without increasing the level of noise, by randomly sampling a subset of the input dataset before applying a DP mechanism; it is commonly used in ML applications. The DP parameters improve proportionally to the subsampling rate (as seen in Theorem 3). Intuitively, each point is less likely to be used in the analysis, and the noise from sampling can be "counted" toward the privacy budget.

For our mechanism, we will focus on *Poisson subsampling*, a sampling process where each element of a population is included in a set according to the outcome of an independent Bernoulli trial; we use $S(q)$ to refer to a Poisson sampling procedure with sampling rate $q$.

**Theorem 3** (Privacy Amplification by Poisson Subsampling (Kasiviswanathan et al., 2010) (Beimel et al., 2013)). *Let $\mathcal{M}$ be an $(\epsilon, \delta)$-DP mechanism, and let $\mathcal{S}(q)$ be a Poisson sampling procedure with sampling rate q. Then $\mathcal{M} \circ \mathcal{S}(q)$ is $(O(\log(q)\epsilon), q\delta)$-DP.*

### A.1  Sampled Generalized Gaussian Mechanism

Next, we present the Sampled Generalized Gaussian Mechanism, SGG, which is a sampled variant of the GG Mechanism. It generalizes the Sampled Gaussian Mechanism (Mironov et al., 2019), a common mechanism in private ML, and it similarly relies on privacy amplification by subsampling (Theorem 3) to attain improved privacy guarantees relative to its non-sampled counterpart. We state SGG in terms of Poisson sampling because the PRV accountant is defined only for Poisson sampling; the mechanism can immediately be extended other types of sampling and the privacy guarantees would still hold under the appropriate accountant.

---

**Algorithm 2** Sampled Generalized Gaussian Mechanism, $SGG_{\beta,\sigma,q}(f, D)$

---

1: **Input** noise parameters $\beta \geq 1$, $\sigma > 0$, sample rate $q \in (0, 1]$ a vector-valued function $f : \mathcal{D} \to \mathbb{R}^d$, database $D \in \mathcal{D}$
2: Compute $l_2$ sensitivity of $f$: $\Delta_2 f = \max_{D,D' \text{ neighbors}} \|f(D) - f(D')\|_2$
3: $S = \emptyset$
4: **for** each data element $x_j \in D$ **do**
5:     With probability $q$, add $x_j$ to $S$
6: **end for**
7: **for** $i = 1$ to $d$ **do**
8:     Sample $Y_i \sim \mathcal{N}_\beta(0, \sigma \cdot \Delta_2 f)$
9: **end for**
10: **Output** $f(S) + (Y_1, \dots Y_d)$

---

The privacy guarantees of the SGG Mechanism follow nearly immediately from privacy of the GG Mechanism (Theorem 3.1), due to privacy amplification by sampling (Theorem 3).

**Theorem 4.** *For any $\beta \geq 1$, $\sigma > 0$, $\delta > 0$, $q \in (0, 1]$, there exists a value $\epsilon$ such that $SGG_{\beta,\sigma,q}(f, D)$ satisfies $(\epsilon, \delta)$-DP for all vector valued $f$ and for all $D$.*

*Proof.* In proof C.2 we show that the GG mechanism is differentially private for any $\beta \geq 1$, $\sigma > 0$ for any function $f$ with bounded sensitivity $\Delta f \geq 0$. Therefore, by the privacy amplification theorem (Theorem 3.1), the GG mechanism applied on a subset $S(q, D) \subseteq D$, where $S$ is generated by Poisson subsampling from dataset $D$ with probability $q$, is also differentially private. □

### A.2  $\beta$-DP-SGD is Differentially Private

In Algorithm 6, we introduced a new algorithm $\beta$-DP-SGD; in this section we show that the algorithm is private, and that is satisfies the same DP guarantees as the application of the SGG mechanism, given the correct parameter assignment. At a high-level, this follows because the $\beta$-DP-SGD algorithm is the SGG mechanism applied repeatedly, which we showed is differentially private in Theorem 4.

The following theorem states that $\beta$-DP-SGD is differentially private

**Theorem 5.** *For any $\delta > 0$, $\beta \geq 1$, $\sigma > 0$, $f : \mathcal{D} \to \mathbb{R}^d$, database $D \in \mathcal{D}$, for any loss function of the form $l(\theta, x_i)$, learning rate $\eta \geq 0$, average group size L, clipping norm $C \geq 0$, there exists $\epsilon \geq 0$ such that algorithm $\beta$-DP-SGD$(\beta, \sigma, D, l, \eta, L, C)$ satisfies $(\epsilon, \delta)$-DP.*

*Proof.* Looking to Algorithm 6, the final output $\theta_T$ can be written as $\theta_T = \theta_0 + \eta \sum_t^T \tilde{G}_t$. As such, the final model only accesses the dataset through the computation of $\tilde{G}_t$, necessitating that if each $\tilde{G}_t$ is differentially private then the final model is as well, through the principle of post-processing.

We next claim that the computation of $\tilde{G}_t \leftarrow \frac{1}{L} \left( \sum_i \bar{G}_t(x_i) + \vec{Y} \right)$ is differentially private. First, we observe that by construction, the contribution of a single element $x_i$ to the function $\bar{G}_t(x_i) \leftarrow G_t(x_i)/\max\left(1, \frac{\|G_t(x_i)\|_2}{C}\right)$ is clipped, and thus the sensitivity of $\tilde{G}_t$ is bounded. Next, we observe that $\tilde{G}_t$ is computed by summing the output of the Gaussian Mechanism on a set of $\bar{G}_t$ values. As the Gaussian Mechanism is differentially private, and $\bar{G}_t(x_i)$ is bounded, $\beta$-DP-SGD is differentially private. $\qquad \square$

The following corollary enables us to use the PRV accountant to provide privacy accounting for $\beta$-DP-SGD.

**Corollary 5.1.** *If the $SGG_{\beta, \sigma, q}(f, D)$ mechanism composed T times on function f with sensitivity $\Delta f$ satisfies $(\epsilon, \delta)$-DP, then for any $L \leq |D|$, $C = \Delta f$, and loss function of the form $l(\theta, x_i)$, the $\beta$-DP-SGD$(\beta, \sigma, D, l, \eta, L, C)$ also satisfies $(\epsilon, \delta)$-DP.*

*Proof.* By Appendix A.2 we know that $\beta$-DP-SGD is differentially private. As argued in the theorem, $\beta$-DP-SGDis the repeated application of the Gaussian Mechanism on a Poisson subsampled set $L_t \subseteq D$. By construction, this is the Sampled Gaussian Mechanism, and as such satisfies the same guarantees. $\qquad \square$

# B PRIVACY ACCOUNTING FOR THE GENERALIZED GAUSSIAN MECHANISM

Underpinning any statement about the privacy-accuracy tradeoff of a particular mechanism is the specific implementation of the privacy accountant used for computing the privacy consumed by the mechanism. At the time of writing, numerical accountants such as the PRV accountant achieve the tightest privacy guarantees, however, as a result closed form solutions for the privacy consumed do not typically exist.

Specifically, Gopi et al. (2021) introduced the algorithm ComposePRV, which efficiently computes the privacy guarantees for the composition of multiple DP mechanisms. It takes as input the CDFs of PRVs $Y_1, \ldots, Y_k$, a mesh size $h$, and truncation parameter $L$, and returns an estimate of the privacy curve for all the mechanisms composed, represented by $\delta(\epsilon)$, enabling the direct computation of $\epsilon$. Given ComposePRV, the PRV accountant can be directly used to accurately compute the privacy loss, and can also be used with Poisson subsampled variants like SGG.

However, when using the PRV accountant one must compute the CDF of the PRV, which is not simple to compute for all differentially private mechanisms. In this work, we get around this by estimating the CDF of the PRV numerically; and while this does introduce error, it also makes the privacy accounting of arbitrary mechanisms possible. In this section we provide error bounds for replacing that function with an empirically computed function; such that $CDF(PRV)$ is replaced with $CDF(Hist(PRV, N, b, A, B))$, where $Hist$ is a histogram over the region $[A, B]$ with $b$ bins, computed by sampling the $PRV$ $N$ times.

## B.1 MULTI-DIMENSIONAL PRVS

As derived in Proposition 1, the PRVs for the single dimensional GG mechanism are $Y = |Z - \mu|^\beta - |Z|^\beta$ and $X = |Z|^\beta - |Z - \mu|^\beta$ where $Z \sim \mathcal{N}_\beta(\sigma = 1, \mu = 0)$. And in our work we sample from this distribution in order to generate our CDFs-of-the-PRVs for the PRV accountant. However, the

PRVs for the multidimensional GG mechanism are [6]

$$Y \sim \log\left(\frac{\exp(-|\vec{t}|^\beta/\sigma)}{\exp(-|\vec{t}-\vec{\mu}|^\beta/\sigma)}\right) \tag{2}$$

As we are sampling along each dimension independently, this can be rewritten:

$$Y \sim \log\prod_i^d\left(\frac{\exp(-|t_i|^\beta/\sigma)}{\exp(-|t_i-\mu|^\beta/\sigma)}\right) \tag{3}$$

$$= \sum_i^d(|t_i-\mu_i|^\beta - |t_i|^\beta)/\sigma \tag{4}$$

### B.2 DIMENSION INDEPENDENCE FOR PRIVACY ACCOUNTING OF THE GENERALIZED GAUSSIAN MECHANISM

One important observation is that $Y = \sum_i^d(|t_i-\mu_i|^\beta - |t_i|^\beta)/\sigma$ can be rewritten as

$$Y = \frac{1}{\sigma}\ell_\beta(|\vec{t}-\vec{\mu}|)^\beta - \ell_\beta(|\vec{t}|)^\beta \tag{5}$$

Now the task is to properly account for the $\vec{\mu}$ term. In order to provide a worst-case guarantee of DP, one must choose $\vec{\mu}$ such that it maximizes difference in probability of $\vec{t}$ versus $\vec{t}+\vec{\mu}$. Generally, we say that $\vec{\mu}$ sits on a ball such that $\ell_2(\vec{\mu}) = 1$. However, we observe we can make an important cancellation when $\vec{\mu}$ is on a unit-ball in $\ell_\beta$ ($\mu \in \{x|\ell_\beta(x) = 1\}$). In this case, any choice of $\vec{\mu}$ will satisfy this maximal-difference constraint, because all points on the $\ell_p$ ball are, by definition, exactly $\ell_p = 1$ 1 away. This is seen because $Y = \ell_\beta(|\vec{t}-\vec{\mu}|)^\beta - \ell_\beta(|\vec{t}|)^\beta$ is equivalent for any $\vec{\mu}$ on the $\ell_\beta$ unit-ball. This includes the one-hot vector $\vec{\mu} = <1, 0, \ldots, 0>$. For this $\vec{\mu}$, we have

$$Y \sim \sum_i^d(|t_i-\mu_i|^\beta - |t_i|^\beta)/\sigma = \tag{6}$$

$$= \left((t_0-1)^\beta - t_0^\beta + 0 + \ldots + 0\right)/\sigma \tag{7}$$

$$Y \sim ((t_0-1)^\beta - t_0^\beta)/\sigma \tag{8}$$

As such, for $\beta$-GG mechanism with sensitivity measured with a $\ell_\beta$ norm, the PRV for the multi-dimensional GG distribution is equivalent to the PRV for a single-dimensional GG distribution, when the sensitivity is $\ell_\beta$.

### B.3 PRIVACY OF THE SAMPLED PRV ACCOUNTANT

As a refresher, we rewrite the core algorithm from (Gopi et al., 2021), ComposePRV, which takes the CDF of multiple PRVs ($Y_i$), and returns an estimate of the composed PRV ($\delta_{\tilde{Y}}(\cdot)$).

---

**Algorithm 3** ComposePRV algorithm (Gopi et al., 2021)

---

1: **Input:** CDFs of PRVs $Y_1, Y_2, \ldots Y_k$, mesh size $h$, truncation parameters $L \in \frac{h}{2} + h\mathbb{Z}^{>0}$
2: **output** PDF of an approximation $\tilde{Y}$ for $Y = \sum_{i=1}^k Y_i$. $\tilde{Y}$ will be supported on $\mu + (h\mathbb{Z} \cap [-L, L])$ for some $\mu \in [0, \frac{h}{2}]$
3: **for** $i = 1$ to $k$ do **do**
4:     $\tilde{Y}_i \leftarrow \text{DiscretizePRV}(Y_i, L, h)$
5: **end for**
6: Compute PDF of $\tilde{Y} = \tilde{Y}_1 \oplus_L \tilde{Y}_2 \oplus_L \ldots \oplus_L \tilde{Y}_k$ by convolving PDFs of $\tilde{Y}_1, \tilde{Y}_2, \ldots, \tilde{Y}_k$ using FFT.
7: Compute $\delta_{\tilde{Y}}(\epsilon) = \mathbb{E}_{\tilde{Y}}\left[\left(1 - e^{\epsilon-\tilde{Y}}\right)_+\right]$ for all $\epsilon \in [0, L]$.
8: Return $\tilde{Y}, \delta_{\tilde{Y}}(\cdot)$.

---

[6]And subsequently $X \sim \log\left(\frac{\exp(-|\vec{t}-\vec{\mu}|^\beta/\sigma)}{\exp(-|\vec{t}|^\beta/\sigma)}\right) = -Y$

We seek to extend the ComposePRV algorithm to work even when the CDF of the PRVs is not known. We do this by sampling from the PRV and generating a histogram as an estimate for the CDF of the PRV. We call this the Sampled PRV accountant. Importantly, the ComposePRV algorithm produces an estimate of the composed privacy curve, along with arbitrarily small error bars; in our work, we extend the error analysis to provide error bounds for when using a sampled CDF as well.

Prior to the proof our theorem B.3 about the correctness of a Sampled PRV accountant, we need to define a few terms. We first define a **sampled random variable** $Z'_{n,L,h}$, which is defined by the PMF $P_{Z,n,L,h}$ generated from sampling from RV $Z$, $n$ times with bins of width $h$, supported over the domain $[-L, L]$. In other words, we sample from $Z$ $n$ times each, generating a PMF, defining a random variable. Importantly, we observe that while the sampled RV $Z'_{n,L,h}$ is itself a random variable, the PMF generated by $Z$ is the output of a random variable, and varies. We use the term *sampled-PRV* to refer to a sampled RV generated from a PRV.

We also borrow the notation of *couple approximation* from (Gopi et al., 2021), as follows.

**Definition 9** (Coupling Approximation)**.** *Given two random variables $Y_1, Y_2$, we write that $|Y_1 - Y_2| \leq_\eta h$ if there exists a coupling between $Y_1, Y_2$ such that $\Pr[|Y_1 - Y_2| > h] \leq \eta$.*

Sampling is by definition a random process, and so it is fundamentally unavoidable that with small probability the sampled-PRV can be very wrong. However, it is straightforward to bound the probability of producing a sampled-PRV that is particularly wrong. And so we can produce a PRV-accountant with an associated error rate, and a probability $\delta'$ of producing an accountant that has larger error, for arbitrarily small $\delta'$ (as a function of the number of samples used to generate the sampled-PRV).

**Theorem 6.** *Given a series of PRVs $Y_i$, and the associated sampled-PRVs $Y'_{i,n,L,h}$ generated for each RV sampling $n$ times and producing a PMF to sample from. Using the sampled PRVs with the ComposePRV algorithm returns the privacy curve of the composed mechanisms, with probability $1 - \delta'$, for $\delta' = 2e^{-2nh^2}$ probability, and the following error bounds:*

$$\delta_{\tilde{Y}}(\epsilon + a + \epsilon_{error}) - \delta_{error} - \eta \leq \delta_Y(\epsilon) \leq \delta_{\tilde{Y}}(\epsilon - a - \epsilon_{error}) + \delta_{error} + \eta \tag{9}$$

*for*

$$h = \frac{\epsilon_{error}}{\sqrt{\frac{k}{2} \log \frac{12}{\delta_{error}}}} \tag{10}$$

*, and $L \geq 2 + \epsilon_{error}$ sufficiently large such that*

$$\sum_{i=1}^{k} \delta_{\mathcal{M}_i}(L - 2) \leq \frac{\delta_{error}}{8} \text{ and } \delta_{\mathcal{M}}(L - 2 - \epsilon_{error}) \leq \frac{\delta_{error}}{4} \tag{11}$$

*for $a := h\sqrt{2k\log\frac{2}{\eta}} + \frac{\bar{\sigma}}{\sqrt{n}}k$, where $\bar{\sigma}$ is the maximum standard deviation of PRV $Y_i$ for all $i$.*

*And our runtime is $O(b\frac{L}{h} \log \frac{L}{h} + nb)$ where $b$ is the number of distinct algorithms amongst $\mathcal{M}_1, \ldots, \mathcal{M}_k$.*

*Proof.* **Outline:** The outline of our proof is that we will establish a coupling between a random variable and the sampled random variable. Then extend a lemma from (Gopi et al., 2021) in order to produce a coupling between a composed sampled random variable, and the random variable. Then we will use the triangle inequality in order to bound the max $\epsilon$ and $\delta$ difference between the composed $\delta(\epsilon)$ and the $\tilde{\delta}$ generated from ComposePRV on a sampled CDF.

**Coupling between PRV and Sampled PRV.** First, we observe that (Dvoretzky et al., 1956) and (Massart, 1990) provide a tight bound for the error produced by a sampled CDF, represented as such:

$$\Pr[sup_{x \in \mathbb{R}} |F_n(x) - F(x)| > \epsilon] \leq 2e^{-2n\epsilon^2} \tag{12}$$

As such, (Massart, 1990) provides a coupling between $Z$ and $Z'_n$, and with probability $1 - \delta'$, $|Z'_{n,L,h} - Z| \leq_0 \epsilon$; this is notably different than $|Z'_{n,L,h} - Z| \leq_{\delta'} \epsilon$. The difference here is that

the probability is over samples drawn to generate the PMF $P_{Z,n,L,h}$, rather than samples of $Z'_{n,L,h}$. Specifically, with probability equal to $\delta' = 2e^{-2n\epsilon_{\text{sample}}^2}$, we see that $|Z - Z'_n| \leq_0 \epsilon_{\text{sample}}$. Here we choose $\epsilon_{\text{sample}} \leq \frac{h}{2}$ from discretization.

Further, we observe that it is possible to define both a coupling between $Z$ and RV $Z'_{n,L,h}$, as well as between $\mathbb{E}[Z]$ and RV $\mathbb{E}[Z'_{n,L,h}]$. We observe that computing a sample mean is unbiased, and that the sample mean distribution $Z'$ has an expected value of $\mathbb{E}[Z]$ and a variance equal to $\frac{\sigma^2}{n}$, where $\sigma$ is the variance of $Z$. Therefore, we are able to bound the probability that $|\mathbb{E}[Z'] - \mathbb{E}[Z]| \geq a$ with chehyshev's inequality. $\Pr[|\mathbb{E}[Z'] - \mathbb{E}[Z]| \geq c \cdot \sigma] \leq \frac{1}{c^2}$. We don't necessarily know the $\sigma$ of the true PRV, but we can upper bound it $\bar{\sigma}$, and thus still say that $\Pr[|\mathbb{E}[Z'] - \mathbb{E}[Z]| \geq c \cdot \sigma] \leq \frac{1}{c^2}$

We also observe that if RV $Y$ and $\tilde{Y}$ are coupled like $|Y - \tilde{Y} \leq_0 h$, then their expected values are also coupled $|\mathbb{E}[Y] - \tilde{\mathbb{Y}}| \leq_0 h$ by linearity of expectations.

**Coupling the composed mechanisms:** We now seek to couple the composed sampled PRV to the composed PRV. We derive a small spin on lemma 5.3 from (Gopi et al., 2021), by enabling the means of the mechanisms to not be equivalent, as follows:

**Lemma 7.** *Suppose $Y_1, Y_2, \ldots, Y_k$ and $\tilde{Y}_1, \tilde{Y}_2, \ldots, \tilde{Y}_k$ are two collections of independent random variables such that $|Y_i - \tilde{Y}_i| \leq_0 h$ and $|\mathbb{E}[Y_i] - \mathbb{E}[\tilde{Y}_i]| \leq_0 \zeta$ for all $i$,*

$$|\sum_i^k Y_i - \sum_i^k Y'_i| \leq_\eta h\sqrt{2k\log\frac{2}{\eta}} + k\zeta \tag{13}$$

*Proof.* Let $X_i = Y_i - \tilde{Y}_i$ where $(Y_i, \tilde{Y}_i)$ are coupled such that $Y_i - \tilde{Y}_i \leq h$ w.p. 1. Then $X_i \in [-h, h]$ w.p. 1. Since we bound the error in the expected values, we know that $\sum_i^h X_i \leq n \cdot \alpha$. Note that $X_1, X_2, \ldots X_k$ are independent of each other.

As a refresher, Hoeffding's inequality is

$$\Pr[\sum_i^n X_i - E[\sum_i^n X_i] \geq t] \leq \exp\left(\frac{2t^2}{\sum_i^k (b_i - a_i)^2}\right) \tag{14}$$

where $a_i \leq X_i \leq b_i$.

Let $S_n = \sum_i^n X_i$ Hoeffding's inequality states that $Pr[|S_n - \mathbb{E}[S_n]| \geq t] \leq 2\exp\left(-\frac{2t^2}{\sum_i^n (b_i - a_i)^2}\right)$. We know that the expectation of $\sum_i^n Y'_i$ is near the expectation of $\sum_i^n Y_i$, so we can add the probability of the distance from the expectation to the right-hand side of the inequality. By application to the Hoeffding's inequality, we can see that

$$Pr[|\sum_i^k X_i| \geq t + (\zeta n)] \leq 2\exp\left(-\frac{2t^2}{k(2h)^2}\right) = \eta \tag{15}$$

$$\text{Let } t := h\sqrt{2klog\frac{2}{\eta}} \tag{16}$$

$$|\sum_i^k Y_i - \sum_i^k Y'_i| \leq_\eta h\sqrt{2klog\frac{2}{\eta}} + k\zeta \tag{17}$$

$\square$

**Connecting the Composed PRV to a Privacy Curve**

We recall Theorem 5.5 from (Gopi et al., 2021), which states

**Theorem 8.** *Let $\epsilon_{error}, \delta_{error} > 0$ be some fixed error terms. Let $\mathcal{M}_1, \mathcal{M}_2, \ldots, \mathcal{M}_k$ be DP algorithms with privacy curves $\delta_{\mathcal{M}_i}(\epsilon)$. Let $Y_i$ be the PRV corresponding to $\mathcal{M}_i$ such that $\delta_{\mathcal{M}_i}(\epsilon) = \delta_{Y_i}(\epsilon)$*

*for $\epsilon \geq 0$. Let $\mathcal{M}$ be the (adaptive) composition of $\mathcal{M}_1, \mathcal{M}_2, \ldots, \mathcal{M}_k$ and let $\delta_{\mathcal{M}}(\epsilon)$ be its privacy curve. Set $L \geq 2 + \epsilon_{error}$ sufficiently large such that*

$$\sum_{i=1}^{k} \delta_{\mathcal{M}_i}(L-2) \leq \frac{\delta_{error}}{8} \text{ and } \delta_{\mathcal{M}}(L-2-\epsilon_{error}) \leq \frac{\delta_{error}}{4} \tag{18}$$

*Let $\tilde{Y}$ be the approximation of $Y = \sum_{i=1}^{k} Y_i$ produced by ComposePRV algorithm with mesh size*

$$h = \frac{\epsilon_{error}}{\sqrt{\frac{k}{2} \log \frac{12}{\delta_{error}}}} \tag{19}$$

*Then*

$$\delta_{\tilde{Y}}(\epsilon + \epsilon_{error}) - \delta_{error} \leq \delta_Y(\epsilon) = \delta_{\mathcal{M}(\epsilon)} \leq \delta_{\tilde{Y}}(\epsilon - \epsilon_{error}) + \delta_{error} \tag{20}$$

*Furthermore, our algorithm takes $O(b\frac{L}{h}\log\frac{L}{h})$ time where $b$ is the number of distinct algorithms amongst $\mathcal{M}_1, \ldots, \mathcal{M}_k$*

Now, we wish to connect the composed PRV to the composed sampled PRV. We recall that by Lemma 5.2 from (Gopi et al., 2021), which states that

**Lemma 9.** *If $Y$ and $\tilde{Y}$ are two random variables such that $|Y - \tilde{Y}| \leq_\eta h$ then, for every $\epsilon \in \mathbb{R}$.*

$$\delta_{\tilde{Y}}(\epsilon + h) - \eta \leq \delta_Y(\epsilon) \leq \delta_{\tilde{Y}}(\epsilon - h) + \eta \tag{21}$$

**Putting it all together:**
We sample $n$ times, and derive PRV $Y'_{i,n,L,h}$ from $Y_i$, which gives us $Z'_{n,L,h}$, such that probability equal to $\delta' = 2e^{-2n\epsilon_{sample}^2}$, we see that $|Z - Z'_{n,L,h}| \leq_0 \epsilon_{sample}$. Which we are able to plug into Lemma 7 in order to bound the max error in the PRV.

Given that the two composed mechanisms $\sum Y_i$ and $\sum Y'_i$ are coupled $|\sum_i^n Y_i - \sum_i^n Y'_i| \leq_\eta h\sqrt{2klog\frac{2}{\eta}} + \frac{1}{c^2}$. Let $a := h\sqrt{2klog\frac{2}{\eta}} + \frac{1}{c^2}$. By Lemma 9, $\delta_{\tilde{Y}}(\epsilon + a) - \eta \leq \delta_Y(\epsilon) \leq \delta_{\tilde{Y}}(\epsilon - a) + \eta$.

Applying the triangle inequality, to include the error from sampling and from discretization, we see that $\delta_{\tilde{Y}}(\epsilon + a + \epsilon_{error}) - \delta_{error} - \eta \leq \delta_Y(\epsilon) \leq \delta_{\tilde{Y}}(\epsilon - a - \epsilon_{error}) + \delta_{error} + \eta$.

In conclusion, sampling $n$ times, to generate RV $Y'_{i,n,L,h}$ provides a $1 - \delta'$, for $\delta' = 2e^{-2nh^2}$ probability of producing an accountant that has error bounds: $\delta_{\tilde{Y}}(\epsilon + a + \epsilon_{error}) - \delta_{error} - \eta \leq \delta_Y(\epsilon) \leq \delta_{\tilde{Y}}(\epsilon - a - \epsilon_{error}) + \delta_{error} + \eta$, and $a := h\sqrt{2klog\frac{2}{\eta}} + \frac{\bar{\sigma}}{\sqrt{n}}k$

The runtime is simple to compute, as composePRV takes $O(b\frac{L}{h}\log\frac{L}{h})$ time where $b$ is the number of distinct algorithms amongst $\mathcal{M}_1, \ldots, \mathcal{M}_k$ (see Theorem 8), and our only addition is the sampling, which takes $n * b$ sampling operations. So, our run time is $O(b\frac{L}{h}\log\frac{L}{h} + nb)$

$\square$

### B.4 MECHANISMS WITH EQUIVALENT PRIVACY GUARANTEES

For any privacy accountant it is generally possible to run the accounting algorithm to compute the hyperparemeters required to achieve a particular degree of privacy. We introduce the following, simple but effective algorithm for using the PRV accountant as part of a binary search over possible values of $\sigma$ in order to compute the minimal $\sigma$ value that satisfied $(\epsilon, \delta)$-DP for a given $\beta$.

Let $PRV(\beta, \sigma_{min}, \delta)$ be a function that runs the PRV accountant for the $(\beta, \sigma)$-GG mechanism, and returns the $\epsilon$ value associated, such that $(\beta, \sigma)$-GG satisfies $(\epsilon, \delta)$-DP.

---

**Algorithm 4** Binary-search $\sigma$-solver

---
1: Input: $\beta \geq 1$, $\epsilon > 0$, $\delta > 0$, tolerance $> 0$
2: Output: $\sigma$, such that $(\beta, \sigma)$-GG satisfies $(\epsilon, \delta)$-DP
3: $\sigma_{min} = \sigma_{max} = 1$
4: **while** $PRV(\beta, \sigma_{min}, \delta) > \epsilon$ **do**
5: $\quad \sigma_{min} = \sigma_{min}/2.$
6: **end while**
7: **while** $PRV(\beta, \sigma_{max}, \delta) < \epsilon$ **do**
8: $\quad \sigma_{max} = \sigma_{max} * 2.$
9: **end while**
10: **while** $PRV(\beta, \sigma_{max}, \delta) - \epsilon >$ tolerance **do**
11: $\quad \sigma_{mid} = \frac{\sigma_{max} + \sigma_{min}}{2}$
12: $\quad$ **if** $PRV(\beta, \sigma_{mid}, \delta) > \epsilon$ **then**
13: $\quad\quad \sigma_{min} = \sigma_{mid}$
14: $\quad$ **else** $\sigma_{max} = \sigma_{mid}$
15: $\quad$ **end if**
16:
17: **end while**
18: **return** $\sigma_{max}$

---

For most of our empirical sections we wish to compare how the choice of $\beta$ changes the accuracy, independent of privacy guarantees. So, a variation on this binary search solver is used to compute which mechanisms to compare in the empirical results presented in the paper for both the private argmax and private DP-SGD sections.

### B.5 OUTLIERS FOR EQUIVALENTLY PRIVATE MECHANISMS

Using the PRV privacy accountant, we are able to solve for $\sigma$ as function of $(\epsilon, \delta, \beta)$, such that $GG_{\beta,\sigma}(f, D)$ satisfies $(\epsilon, \delta)$-DP (described in Appendix B.4). At the present this is not possible with other privacy accountants, and it is certainly not possible for arbitrary private mechanism.

Combining this empirical privacy accountant with the known CDF of the GG distribution (Dytso et al., 2018), we can compute the weight of the tail, as a function of $(\beta, \epsilon, \delta, d)$, where $d$ specifies what we defines as a tail (the cutoff point for what constitutes an outlier). Appendix B.5 clearly shows that there are regimes where the tails of Laplace and Gaussian are heavier (outliers are more likely) than other equivalently private mechanisms (for $\beta \notin \{1, 2\}$).

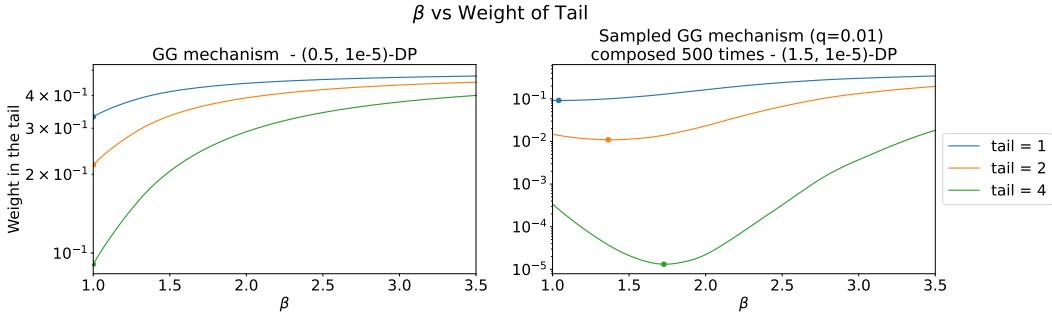

Figure 5: Likelihood of outliers for the GG mechanism and sampled GG mechanism. Here, $q$ is the Poisson sampling probability used in the Sampled Generalized Gaussian Mechanism.

This observation provides a potential direction for how to evaluate and search for alternative DP mechanisms; we observe that minimizing outliers is one of the main considerations cited by both the US Census (Abowd et al., 2022) and researchers behind PATE (Papernot et al., 2018) when deciding to use the Gaussian mechanism instead of the Laplace mechanism. However, more work needs to be done in this area, as in general, an algorithm designer's goal is not to minimize outliers, but rather to minimize some loss function (e.g. maximize accuracy).

## C  OMITTED PROOFS

### C.1  RÉNYI DIVERGENCE OF THE GENERALIZED GAUSSIAN IS BOUNDED

**Lemma 10.** $\forall \alpha > 1, \forall \beta \geq 1, \forall \sigma > 0, \forall \mu > 0,$ *the Renyi Divergence* $D_\alpha(GG(\beta, \sigma, 0) \| GG(\beta, \sigma, \mu))$ *is bounded.*

*Proof.*

$$D_\alpha(P \| Q) = \frac{1}{\alpha - 1} \log \mathbb{E}_{x \sim Q} \left[ (P(x)/Q(x))^\alpha \right]$$

$$\text{thus } D_\alpha(GG(\beta, \sigma, 0) \| GG(\beta, \sigma, \mu)) = \frac{1}{\alpha - 1} \log \int_{-\infty}^{\infty} k \cdot \exp(-\frac{\alpha}{\sigma} |x|^\beta) \cdot \exp(-\frac{(1 - \alpha)}{\sigma} |x - \mu|^\beta) dx$$

As such, the goal is to bound $\int_{-\infty}^{\infty} \exp \left( -\frac{\alpha}{\sigma} \||x - \mu|\|^\beta - \frac{1 - \alpha}{\sigma} \||x - \mu|\|^\beta \right) dx$

The shape of the proof is as follows:

1. We show that the there is a fixed region $[a, b]$ over which the sign of the exponent is positive, and outside of which it is negative.

2. We therefore are able to bound the integral of the exponential over the region $[a, b]$, as $(b - a) \cdot \max$, where $\max$ is the maximum value of the integral (which is finite, because it is concave down.)

3. We then show that for the regions outside of $[a, b]$, the exponential decays faster than a Laplace distribution, which has a bounded integral, therefore, the region outside of $[a, b]$ also has a bounded integral.

The power of the exponential is $-\frac{\alpha}{\sigma} \left( |x|^\beta - \frac{\alpha - 1}{\alpha} |x - \mu|^\beta \right)$, which we denote by $R_{(\beta, \sigma)}(x)$ for ease of reference. We observe that its roots are the solution to $\left( \frac{|x|}{|x - \mu|} \right)^\beta = \frac{(\alpha - 1)}{\alpha}^{1/\beta}$.

There are only two roots to function, which we refer to as $(a, b)$.

$$\left( \frac{|x|}{|x - \mu|} \right) = \frac{(\alpha - 1)^{1/\beta}}{\alpha}$$

$$\left( \frac{x^2}{(x - \mu)^2} \right)^{1/2} = \frac{(\alpha - 1)^{1/\beta}}{\alpha}$$

$$x^2 = \frac{(\alpha - 1)^{2/\beta}}{\alpha} (x - \mu)^2$$

$$0 = \frac{(\alpha - 1)^{2/\beta}}{\alpha} (x - \mu)^2 - x^2$$

We note that the roots $a, b$ do not depend on $\sigma$. For more intuition, we plot $\frac{|x|}{|x - \mu|}$ below in Figure 6.

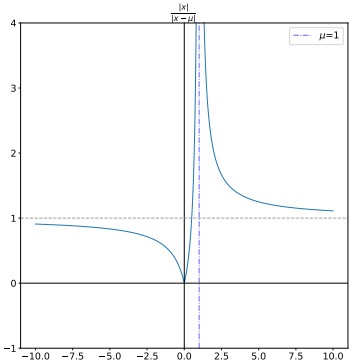

Figure 6: A plot of $\frac{|x|}{|x-\mu|}$ for $\mu = 1$.

The curve $R_{(\beta,\sigma)}(x)$ is bounded in the region $[a, b]$, because it is a polynomial. Thus, over this region, it must attain some maximum value denoted $m$. This means that the integral $\int_a^b \exp[R_{(\beta,\sigma)}(x)]dx$ is bounded by some value $(b - a) \cdot m$.

Next, we show that the curve $R_{(\beta,\sigma)}(x)$ is always negative in the region $(-\infty, a) \cup (b, \infty)$. The remainder of our proof goes as follows,

1. First we show that there is an $p$ such that $x > p$, the sign of the exponent is negative.

2. Then we show that there is a negative value $n$ such that for $x < n$ the sign of the exponent is negative.

3. Taking the two previous points, combined with the fact that there are only two roots (at $a$ and $b$), then the sign of the exponent is negative over the region $(-\infty, a) \cup (b, \infty)$.

Let $f_\mu(x) := \frac{|x|}{|x-\mu|}$. First, we observe that if $f_\mu(x) > \frac{\alpha-1}{\alpha}$ the sign of the exponent is negative.

**There exists $p$ such that for all $x > p$ the sign of the exponent is negative.**

First, we observe that $\frac{|x|}{|x-\mu|} = 1$ if and only if $\mu = 0$, and that $\frac{\alpha-1}{\alpha} < 1$, for all $\alpha > 1$, thus ${\frac{\alpha-1}{\alpha}}^{1/\beta} < 1$. With this, we next observe that for $x > \frac{\mu}{2}$, $\frac{|x|}{|x-\mu|} > 1$. This implies $\frac{|x|}{|x-\mu|} > {\frac{\alpha-1}{\alpha}}^{1/\beta}$. Thus, there exists somve value $p$ such that for all $x > p$ the sign of the exponent is negative.

**There exists $n$ such that for all $x < n$ the sign of the exponent is negative.**
for $x < 0$, $f_\mu(x) = \frac{|x|}{|\mu-x|} = \frac{-x}{\mu-x}$.

$$\frac{d}{dx}\frac{-x}{\mu-x} = -1 * \left(\frac{1}{\mu-x} + \frac{x}{(\mu-x)^2}\right)$$
$$= \left(1 - \frac{x}{\mu-x}\right)\frac{1}{\mu-x}$$

We evaluate this derivative for $x < 0$, and find that $\frac{1}{\mu-x}$ is always positive, whereas $\left(1 - \frac{x}{\mu-x}\right)$ is always negative . Thus, the derivative over the region $x < 0$ is always negative.

This means that given that $f(a) = {\frac{\alpha-1}{\alpha}}^{1/\beta}$, then for all $x < a$, $f(a) > {\frac{\alpha-1}{\alpha}}^{1/\beta}$. Thus, as x goes to negative infinity, the sign of the exponent is negative.

We have so far shown that for $x \in (-\infty, a) \cup (b, \infty)$, the power of the exponent is always negative. We observe that $\exp[R_{(\beta,\sigma)}(x)]$ decays at least as fast as the inverse exponential function $c \cdot \exp(-|x|/d)$, for some value of $(c, d)$, because $\beta \geq 1$.

Given that the infinite integral of the inverse exponential function is bounded (by $c \cdot d$), we can see that there exist values $(c, d)$, such that the values of $\exp[R(x)]$ are pointwise dominated by over the region $(-\infty, a) \cup (b, \infty)$. Therefore, we have shown that the integral over this region is also bounded.

Thus, we have shown that this integral is bounded, and therefore the Rényi divergence for the $(\beta, \sigma)$ distribution is also bounded.

$\square$

### C.2 Generalized Gaussian is Differentially Private

In order to prove Theorem 3.1, we first show that for all $\mu \geq 0$ (corresponding to the difference in the value of $f$ on neighboring databases) and all $\alpha > 1$, the $\alpha$-Rényi divergence between $\mathcal{N}_\beta(0, \sigma)$ and $\mathcal{N}_\beta(\mu, \sigma)$ is bounded by some finite $r$; this step is shown formally in Appendix C.1. Bounded Rényi divergence means that the $GG_{\beta,\sigma}(f, D)$ mechanism satisfies $(\alpha, r)$-RDP, and by the RDP-to-DP conversion of Mironov (2017), it also satisfies $(r + \frac{\log 1/\delta}{\alpha - 1}, \delta)$-DP for any $\delta \in (0, 1)$.

**Theorem (Restatement of Theorem 3.1).** *For any dataset $D \subset \mathbb{R}^{n \times d}$, and any function $f : \mathbb{R}^{n \times d} \to \mathbb{R}$, with sensitivity $\Delta f \geq 0$, then $\forall \beta \geq 1, \forall \sigma > 0$, the $GG_{\beta,\sigma}(f, D)$ is differentially private.*

*Proof.* By Appendix C.1 we know that the Rényi divergence of the associated GG distribution is bounded for $GG(\beta, \sigma, 0) \| GG(\beta, \sigma, \Delta f)$. By the definition of Rényi Differential Privacy (Mironov, 2017), restated in Definition 6, we know that a randomized mechanism that for any adjacent $D, D'$ such that it holds that the Rényi Divergence with order $\alpha$ is less than $\epsilon$ satisfies $(\alpha, \epsilon)$ RDP. Thus we know that the GG mechanism satisfies RDP, and subsequently satisfies DP, as for any RDP bound there is a $(\epsilon, \delta)$-DP guarantee (stated and proved in Mironov (2017)). $\square$

### C.3 Analytic PRV for a single dimension Generalized Gaussian Mechanism

Since, we already know the CDF of the GGD, which is $\frac{1}{2} + sign(x - \mu) \cdot \frac{1}{2\Gamma(1/\beta)} \cdot \gamma\left(1/\beta, \frac{|x-\mu|^\beta}{\alpha}\right)$ (Dytso et al., 2018). Then we observe that for $\beta > 1$ the function $g(x) = |x|^\beta - |x - \mu|^\beta$ is invertible. For $\beta > 1$, $g(x)$ is a monotonic function which a well-defined 1st derivative, and so can be easily computed to arbitrary precision using binary search.

Thus we can compute the CDF with arbitrary precision, as such $F_Y(y) = Pr[|Z - \mu|^\beta - |Z|^\beta \leq y] = Pr[g(Z) \leq y] = Pr[Z \leq g^{-1}(y)] = F_Z(g^{-1}(y))$. Since both $g^{-1}$ and $F_Z$ are known, the PRV is also known.

In short, in order to use the PRV accountant, one must compute the CDF of the two PRVs $X$ and $Y$, the CDF of the $Y$ PRV is $F_Z(g^{-1}(y))$, where $Z \sim GGM(0, c)$ and $g(x) = |x|^\beta - |x - \mu|^\beta$, and $X$ PRV is $F_Z(h^{-1}(y))$, where $h(x) = -g(x)$.

### C.4 Privacy Loss Random Variables for Generalized Gaussian Mechanism

In order to compute the PRV for $GG_{\beta,\sigma}(f, D)$ we consider the privacy loss random variables for two distributions shifted by $\mu = \Delta f$, corresponding to the outputs of the mechanism on two neighboring datasets: $P \sim \mathcal{N}_\beta(0, \sigma)$ and $Q \sim \mathcal{N}_\beta(\mu, \sigma)$.

**Proposition 1.** *Let $Z \sim \mathcal{N}_\beta(0, \sigma)$, and let $\mu = \Delta f$. Then the PRVs for $GG_{\beta,\sigma}(f, D)$ are $X = \left(\frac{1}{\sigma}\right)^\beta \left(|Z|^\beta - |Z - \mu|^\beta\right)$ and $Y = \left(\frac{1}{\sigma}\right)^\beta \left(|Z - \mu|^\beta - |Z|^\beta\right)$.*

*Proof.*

$$Y \sim \log\left(\left(\frac{Q(t)}{P(t)}\right) = \log\left(\frac{\exp(-t^\beta)}{-|t - \mu|^\beta}\right) = |t - \mu|^\beta - |t|^\beta, \text{ where } t \sim Q = \mathcal{N}_\beta(0, 1)\right.$$
$$= |Z - \mu|^\beta - |Z|^\beta, \text{ where } Z \sim \mathcal{N}_\beta(0, 1)$$

A similar calculation shows that $X = |Z|^\beta - |Z - \mu|^\beta$ where $Z \sim GG(\beta, 1)$. $\square$

## C.5 GGNMAX IS PRIVATE

**Theorem** (**Restatement of Theorem 2**). *If the* $(\beta, \sigma)$*-Generalized Gaussian Mechanism is* $(\epsilon, \delta)$*-DP for a fixed* $\epsilon > 0$ *and* $\delta \geq 0$*, then* $(\beta, \sigma)$*-Generalized Gaussian Private Argmax is also* $(\epsilon, \delta)$*-DP.*

The proof of Theorem 2 follows closely to the proof of privacy of the ReportNoisyMax algorithm with Laplace noise, as presented in Dwork & Roth (2014). The proof is included here for completeness.

*Proof.* Let $Z \sim \mathcal{N}_\beta(0, \sigma)$ be the noise sampled in the $GG_{\beta,\sigma}(f, D)$ mechanism, where the $\Delta f$ is equivalent for the Mechanism's $f$ and the Argmax(in most regimes $\Delta f = 1$ for histograms).

Fix $D = D' \cup \{a\}$. Let $c$, respectively $c'$ denote the vector of counts when the database is $D$, respectively $D'$. We use two properties:

1. *Monotonicity of counts.* For all $j \in [m]$, $c_j \geq c'_j$

2. *Lipschitz property.* For all $j \in [m]$, $1 + c'_j \geq c_j$

Fix any $i \in [m]$. We will bound from above and below ratio of the probabilities that $i$ is selected with $D$ and with $D'$. Fix $r_{-i}$, a draw from $[\mathcal{N}_{\beta,\sigma}]^{m-1}$ used for all the noisy counts except the $i$th count. We use the notation $Pr[i|\zeta]$ to mean the probability that the output of the GGNmax algorithm is $i$ conditioned on $\zeta$.

We first argue that $\Pr[i|D, r_{-i}] \leq e^\epsilon \Pr[i|D', r_{-i}] + \delta$. Define

$$x^* = \min_{x_i} : c_i + x_i > c_j + x_j \forall j \neq i$$

Note that, having fixed $x_{-i}$, $i$ is will be the output (the GGNmax noisy count) when the database is $D$ if and only if $x_i \geq x^*$. We have, for all $1 \leq j \neq i \leq m$:

$$c_i + x^* > c_j + x_j$$
$$\Rightarrow (1 + c'_j) + x^* \geq c_i + x^* > c_j + x_j \geq x'_j + r_j$$
$$\Rightarrow c'_i + (x^* + 1) > c'_j + x_j$$

Thus, if $x_i \geq x^* + 1$, then $i$ will be the output (the GGNmax noisy count) on database $D$ with randomness $(r_i, x_{-i})$. Thus, if $r_i \geq x^* + 1$, then the $i$th count will be the maximum when the database is $D'$ and the noise vector is $(x_i, x_{-i})$. So, we now wish to compare (and bound) the probability of $\Pr[x_i \geq 1 + x^*]$ to the probability $\Pr[x_i \geq x^*]$.[7]

As a premise, we know that $GG_{\beta,\sigma}(f, D)$ is $(\epsilon, \delta)$-DP, which states that $Pr[i|D] \leq e^\epsilon \Pr[i|D'] + \delta$ for all neighboring $(D, D')$. Leading us to the following:

$$\Pr[Z \geq x^*] \leq \Pr[Z \geq x^* + \Delta f] + \delta$$
$$\Rightarrow \Pr[i|D, x_{-i}] = \Pr[x_i \geq x^*] \leq e^\epsilon \Pr[x_i \geq x^* + \Delta f] + \delta \leq e^\epsilon \Pr[i|D', x_{-i}] + \delta$$

We now argue that $Pr[i|D'] \leq e^\epsilon \Pr[i|D] + \delta$, which goes in a similar fashion. Having fixed $x_{-i}$, $i$ will be the output (GGNmax noisy count) when the database is $D'$ if and only if $x_i \geq x^*$. Define, again,

$$x^* = \min_{x_i} : c'_i + x_i > c'_j + x_j \forall j \neq i$$

Note that, having fixed $x_{-i}$, $i$ is will be the output (the GGNmax noisy count) when the database is $D'$ if and only if $x_i \geq x^*$. We have, for all $1 \leq j \neq i \leq m$:

$$c'_i + x^* > c'_j + x_j$$
$$\Rightarrow 1 + c'_i + x^* > 1 + c'_j + x_j c$$
$$\Rightarrow c'_i + (1 + x^*) > (1 + c'_j) + x_j$$
$$\Rightarrow c'_i + (1 + x^*) \geq c'_i + (x^* + 1) > (1 + c'_j) + x_j \geq c_j + x_j$$

---

[7]This is the critical part that diverges from the Report Noisy Max proof

Thus, if $x_i \geq x^* + 1$, then $i$ will be the output (the GGNmax noisy count) on database $D$ with randomness $(x_i, x_{-i})$. Once again we a similar equation:

$$\Pr[Z \geq x^*] \leq \Pr[Z \geq x^* + \Delta f] + \delta$$
$$\Rightarrow \Pr[i|D', x_{-i}] = \Pr[x_i \geq x^*] \leq e^\epsilon \Pr[x_i \geq x^* + \Delta f] + \delta \leq e^\epsilon \Pr[i|D, x_{-i}] + \delta$$

And so, we have proven the second direction necessary for DP: $Pr[i|D'] \leq e^\epsilon \Pr[i|D] + \delta$, proving that the GGNmax mechanism satisfies the same guarantee as the 1-dimensional $GG_{\beta,\sigma}(f, D)$ mechanism with equivalent sensitivity. $\qquad\square$

## C.6 Generalized Gaussian Mechanism Privacy Proof

In Appendix C.2 we show that the GG mechanism is private, but only provide a very loose bound, because we show that this is computatable using an adaptation to the PRV accountant. We rely on the PRV accountant because by using sampled PRVs we are able to extend the PRV accountant to arbitrary privacy mechanisms with minimal overhead, and because it has been shown to provide empirically tighter guarantees than the RDP accountant. While not directly useful for our applications, we are still able to directly provide an $(\epsilon, \delta)$-DP guarantee for the GG mechanism, though with coefficients that provide minimal interpretability.

**Theorem 11.** *For* $\forall \epsilon > 0, \delta \in [0, 1], \beta \geq 1$, *for all functions* $g_0(\beta, c) : \mathbb{R}^2 \to \mathbb{R}$, *for all* $c > 0$ *such that*

$$\frac{1}{c} \cdot \exp[-c \cdot \frac{-r}{s-1}^\beta] \leq \delta \cdot (\frac{1-s}{r})^{1-\beta},$$

*for* $r = \frac{g_0 \cdot \frac{\epsilon}{c}^{1/\beta}}{(g_0^{\beta/(\beta-1)} + 1)^{(\beta-1)/\beta}}$; $s = \frac{1}{(g_0^{\beta/(\beta-1)} + 1)^{(\beta-1)/\beta}}$, *the mechanism* $GG(\beta, c)$ *is* $(\epsilon, \delta)$-*DP.*

*Proof.* Given an algorithm $M$ and two neighboring datasets $D, D'$, let $f(y)$ be defined as

$$f(y) = \log \frac{\Pr[M(D) = y]}{\Pr[M(D') = y]}$$

for every possible output in the universe of outcomes $y \in \Omega$. The privacy loss random variable (PLRV) is defined as $Z := f(M(D))$. We know that if $\Pr[Z > \epsilon] \leq \delta$ then $M$ is $(\epsilon, \delta)$-DP Canonne et al. (2020).

For a given $\beta$, we compute the domain over which the PLRV exceeds $e^\epsilon$, and then compute the probability of sampling from that domain.

$$\frac{\exp[-c|x|^\beta]}{\exp[-c|x-\mu|^\beta]} \geq e^\epsilon$$
$$\exp[-c(|x|^\beta - |x-\mu|^\beta)] \geq e^\epsilon$$
$$|x-\mu|^\beta - |x|^\beta \geq \frac{\epsilon}{c}$$

In order to evaluate the monotonicity of the above function we take the derivative:

$$t(x) = |x-\mu|^\beta - |x|^\beta = (x+\mu)^{2^{\beta/2}} - (x^2)^{\beta/2};$$
$$t'(x) = -\beta x |x|^{(-2+\beta)} + \beta(x-\mu)|x-\mu|^{(-2+\beta)}.$$

We observe that for all $\beta \geq 1, \mu > 0, t'(x) < 0$, and so $t(x)$ is monotonically decreasing.[8]

Given the above inequality and the constraint $\beta \geq 1$, we know that the inequality can only be satisfied over the region $x \in (-\infty, \frac{\mu}{2})$. However, we are interested in a tighter bound $x \in (-\infty, z)$, for $z \leq \mu/2$.

First, we restate our inequality:

$$|x-\mu|^\beta - |x|^\beta \geq \frac{\epsilon}{c}$$

$$|x-\mu| \geq ((\frac{\epsilon}{c}^{1/\beta})^\beta + |x|^\beta)^{1/\beta}$$

---

[8]It is worth noting that that $t(x)$'s second derivative flips signs at $\beta = 2$, and provides a potential insight into why the Gaussian mechanism is interesting.

**Lemma 12** (Holder's Inequality). *Let $(S, \Sigma, \mu)$ be a measure space and let $p, q \in [1, \infty]$ with $1/p + 1/q = 1$. Then for all measurable real- or complex-valued functions $f$ and $g$ on $S$,*

$$||fg||_1 \leq ||f||_p ||g||_q.$$

We choose $\vec{f}$ and $\vec{g}$ to be the real-valued function of length 2, such that $\vec{f} = < c^{1/\beta}, |x| >$, and $\vec{g} = < g_0, 1 >$, where $g_0$ is not yet specified.

We choose $p, q$ such that $p = \beta$, $q = \frac{\beta}{\beta - 1}$.

By Hölder's inequality,

$$\left( g_0 \cdot \frac{\epsilon^{1/\beta}}{c} + 1 \cdot |x| \right) \leq \left( (c^{1/\beta})^\beta + |x|^\beta \right)^{1/\beta} \cdot (g_0^q + 1)^{1/q}.$$

$$\left( (\frac{\epsilon^{1/\beta}}{c})^\beta + |x|^\beta \right)^{1/\beta} \geq \frac{(g_0 \cdot \frac{\epsilon^{1/\beta}}{c} + |x|)}{(g_0^q + 1)^{1/q}}$$

So

$$|x - \mu| \geq ((\frac{\epsilon^{1/\beta}}{c})^\beta + |x|^\beta)^{1/\beta} \geq \frac{(g_0 \cdot \frac{\epsilon^{1/\beta}}{c} + |x|)}{(g_0^q + 1)^{1/q}}$$

for algebraic ease, let $r := \frac{g_0 \cdot \frac{\epsilon^{1/\beta}}{c}}{(g_0^q + 1)^{1/q}}$, and $s := \frac{1}{(g_0^q + 1)^{1/q}}$. We now have

$$|x - \mu| \geq r + s|x|.$$

Our goal is to solve for a tight bound for $z$, where $x \in (-\infty, z)$. Given $\mu > 0$, $r > 0$, and $0 < s < 1$, we observe that

$$x \leq \frac{r}{s - 1}.$$

This leads us to $x \in (-\infty, \frac{r}{s-1})$. We note that this holds for all $g_0$, and if we wanted to, we may choose $g_0$ to provide the tightest bounds on x.

In pure DP we show that $\forall y \in Y \Pr(\mathcal{M}(x) = y)$. In order to prove approximate-DP, it is enough to prove that if we draw $y$ from $\mathcal{M}(x)$ then with probability $(1 - \delta)$ we will have $\Pr(\mathcal{M}(x) = y) \leq e^\epsilon \Pr(\mathcal{M}(x') = y)$.

**Now, we bound the probability of drawing $y$ outside of the support of** $\{y | \Pr(\mathcal{M}(x) = y) \leq e^\epsilon \Pr(\mathcal{M}(x') = y)\}$

Using the above inequality, we solve for the region that does not satisfy $(\epsilon, 0)$-DP. Note: if no value of $x$ satisfies this inequality, then this actually satsifies $(\epsilon, 0)$-DP (Pure DP).

Given the region $A := (a, \infty)$ such that $f(x) \geq e^\epsilon$ for $x \in A$. We compute the probability of $x \sim GG(\beta, c) \in A$. This is equivalent to the integral of the PDF over the region $A$. If we are able to upper bound the weight with $\delta$, then we have shown $(\epsilon, \delta)$-DP

$$\int_a^\infty \exp[-c|x|^\beta]dx \leq \int_a^\infty (\frac{x}{a})^{\beta - 1} \cdot \beta \cdot \exp[-c|x|^\beta]dx \tag{22}$$

$$= (\frac{1}{a})^{\beta - 1} (\frac{1}{c} \exp[-c \cdot x^\beta]) \Big|_a^\infty \tag{23}$$

$$= (\frac{1}{a})^{\beta - 1} \cdot \frac{1}{c} \cdot \exp[-c \cdot a^\beta] \tag{24}$$

If $(\frac{1}{a})^{\beta - 1} \cdot \frac{1}{c} \cdot \exp[-c \cdot a^\beta] \leq \delta$, then $M$ satisfies $(\epsilon, \delta)$-DP.

We now combining the two parts of the proof in order to derive an analytic bound.

$$\text{let } r := \frac{g_0 \cdot \frac{\epsilon^{1/\beta}}{c}}{(g_0^{\frac{\beta}{\beta-1}} + 1)^{\frac{\beta-1}{\beta}}}$$

$$\text{let } s := \frac{1}{(g_0^{\frac{\beta}{\beta-1}} + 1)^{\frac{\beta-1}{\beta}}}$$

We observe that the Generalized Gaussian is symmetric around the origin, so we use Equation (22) $a = -\frac{r}{s-1}$. For $\forall \epsilon > 0, \delta \in [0, 1], \beta \geq 1$, for all functions $g_0(\beta, c) : \mathbb{R}^2 \to \mathbb{R}$, for all $c > 0$ that satisfy the following inequality, are $(\epsilon, \delta)$-DP.

$$(\frac{1-s}{r})^{\beta-1} \cdot \frac{1}{c} \cdot \exp[-c \cdot \frac{-r}{s-1}^{\beta}] \leq \delta$$

$$\frac{1}{c} \cdot \exp[-c \cdot \frac{-r}{s-1}^{\beta}] \leq \delta \cdot (\frac{1-s}{r})^{1-\beta}$$

$\square$

## D  FURTHER TREATMENT OF PRIVATE ARGMAX AND PATE

### D.1  GGNMAX ALGORITHM

Below we present the GGNMax Mechanism is algorithmic form.

---
**Algorithm 5** Generalized Gaussian Private Argmax, GGNMax$(\beta, \sigma, \{f\}, \Delta, D)$

---
1: **Input** noise parameters $\beta \geq 1$, $\sigma > 0$, functions $f_1, \ldots, f_N : \mathcal{D} \to \mathbb{R}$ each of sensitivity $\Delta$, database $D \in \mathcal{D}$
2: **for** $i = 1$ to $N$ **do**
3:     Compute $f_i(D)$
4:     Sample $Y_i \sim \mathcal{N}_\beta(0, \sigma\Delta)$
5: **end for**
6: **Output** $\arg\max_{i \in [N]}\{f_i(D) + Y_i\}$

---

### D.2  DESCRIPTION OF PATE

PATE (Private Aggregation of Teacher Ensembles) is an algorithm to train a private machine learning model. In the first step, the private dataset is partitioned into $T$ datasets, such that a single user's data is only in single partition. A "teacher" model is trained for each partition. Then, the teacher models are collected to privately vote on how to label an unlabeled, public dataset, usually through an algorithm based off of the Report-Noisy-Max algorithm. A "student" model is trained on the privately labeled dataset. We present a high-level description of the algorithm in Figure 7.

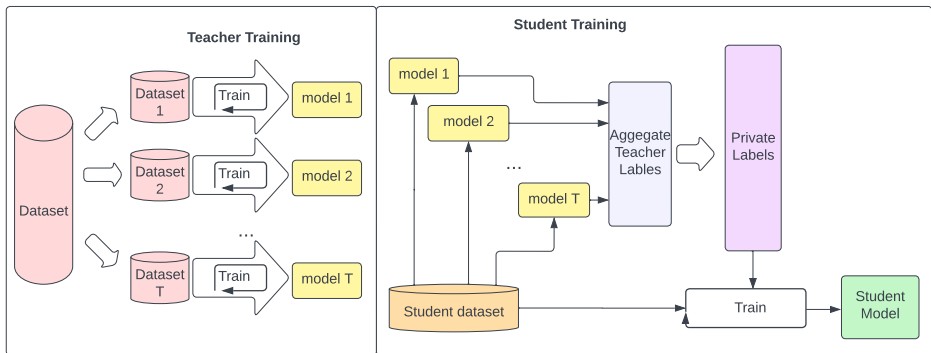

Figure 7: Diagram of PATE implementation laid out in (Papernot et al., 2017)

In the main body of the paper we primarily explore the idea of how the choice of $\beta$ affects the label accuracy of PATE; in order to explore this more fully, we propose a new measure for accuracy for the private Argmax problem, which is better suited to the goals of private ML — measuring the probability of returning the true Argmax, rather than returning an outcome with score similar to the true Argmax. For this utility measure, we empirically find that $\beta = 2$ (Gaussian) is near-optimal. We then explore this in a theoretical setting for the specific case of Laplace and Gaussian mechanism.

### D.3 SIMULATIONS FOR ENSEMBLE-BASED PRIVATE VOTE AGGREGATION

While PATE is one of the primary motivations for the GGNMax mechanism, taking a private Argmax is a very general problem and is particularly important for algorithms which attempt to reconcile beliefs (or votes) across many parties. Classical work in differential privacy on private Argmax considers that for a vector of values, the utility of the mechanism is a function of the probability that the mechanism returns any index that has a value associated with, *close* to the maximum value. However, in a task like classification in ML, there is only a single label that is the "correct" label, and thus we argue that for a task like classification in ML, a mechanism should be evaluated on how often it returns the label that would have been assigned without noise.

Building upon this intuition, we define the *Hardmax Utility* of an Argmax mechanism $\mathcal{M}$ on functions $\{f_i\}$ over a distribution $\mathcal{P}$ of databases as:

$$\text{Hardmax-Utility}_{\mathcal{P}}(\mathcal{M}, \{f_i\}) := \Pr_{D \sim \mathcal{P}}[\mathcal{M}(D, \{f_i\}) = \arg\max_i(f_i(D))].$$

With this utility measure in mind, we wish to measure the impact of $\beta$ on the utility of GG Private Argmax algorithm. Given a vector of function values $\{f_i(D)\}$, noise addition can only change the Argmax if the noise added is larger than the existing gap between the highest function value and all other values. We refer to the difference between the largest and second largest value in $\{f_i(D)\}$, as the *runner-up-gap*.

Our intuition about the effect of $\beta$ is that potential gains in utility would come from how varying $\beta$ will vary the weight of the distributions in the tails, for equivalently private mechanisms (which we explore lightly in Appendix B.5). In order to study this effect, we construct a set of random histograms that have different running-up-gap; by varying the runner-up-gap, we vary how much the outcome is sensitive to outliers in the noise.

In designing our simulated histograms, we observe that it is not clear how to assign ground truth to simulated histograms. However, we also observe that in many ensemble learning based settings, it is reasonable to assume that the majority of the voters are correct. As such, for our simulated histograms, we assign a ground truth label where the argmax of the histogram is the correct label; in other words, we are restricted to the regime where the correct label is what the majority voted for non-privately.

For our simulations, we construct the 500 histograms of votes for each class as follows: for vote count $V = 1000$, maximum value $v = 100$, and runner-up-gaps $r \in [.001, 0.2]$ we fix the largest number of votes for class 0 at $x_0 = v$, the second-largest number of votes for class 1 at $x_1 = v(1 - r)$, and then filling in the histogram by repeatedly drawing $N - 2$ random integers from the range $[1, v(1 - r)]$ until $\sum_{i \in [N]}(\vec{x}_i) = V$. We then instantiate GGNmax on each database (histogram) with counting queries $f_i$ that output the number of votes for each class $i$.

For a given $(\epsilon, \delta)$-DP guarantee and for a range of values $\beta_i \in [1, 4]$, we compute $\sigma_i$, such that $(\beta_i, \sigma_i)$-Generalized Gaussian Private Argmax satisfies $(\epsilon, \delta)$-DP (see Appendix B.4 for algorithmic details of this process). For each pair $(\beta_i, \sigma_i)$, we compute the Hardmax Utility of each mechanism by computing the likelihood of returning the true argmax, after using the $(\beta_i, \sigma_i)$-Generalized Gaussian Private Argmax, averaged across 50 trials.

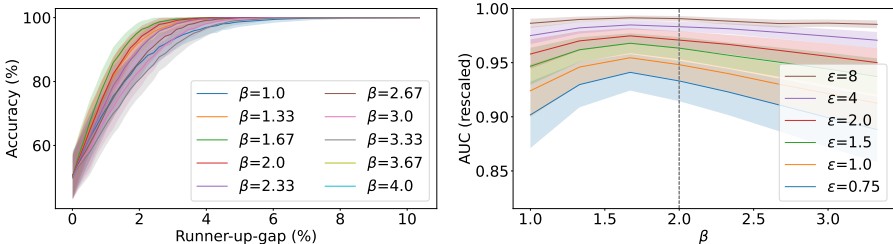

Figure 8: Hardmax Utility for 2-class histogram. Left: Hardmax Utility as a function of runner-up-gap, for mechanisms satisfying $(1, 10^{-5})$-DP. Right: Area-Under-the-Curve (AUC) of the curves on the left for different values of $\epsilon$, where a higher AUC means a better overall accuracy. The AUC is computed over a runner-up-gap % from 0 to 10, such that all $\beta$-GGNmaxes achieve nearly greater than 99.5% accuracy, and then AUC is rescaled such that 1.0 is the maximum.

In Figure 8, we observe that the relative differences across different choices of $\beta$ do not change very much across runner-up-gaps, and the optimal $\beta$ value is typically independent of the runner-up-gap.

We observe that empirically, values of $\beta$ close to $\beta = 2$ have better Hardmax-Utility than ones further away, regardless of $(\epsilon, \delta)$-DP guarantees. However, we do note that while $\beta = 2$ is near-optimal, $\beta$ values slightly smaller than 2 appear to give even better performance. A key takeaway is that Gaussian does outperform Laplace, however, there is room for further improvements over Gaussian by fine-tuning the $\beta$ parameter.

This suggests, empirically, is that for regimes where younpinformal have a set of voters, where there is reason to believe that the majority is generally correct, GGNMax mechanisms with $\beta$ values close to $\beta = 2$ (Gaussian mechanism) may provide the best privacy-accuracy trade-off.

To test our simulation results more generally, we investigate the multi-class regime as well, and find the same results. In Figure 9 we present the same experiments from the previous section Section 4.2 recreated with 25 classes, rather than 2 classes.

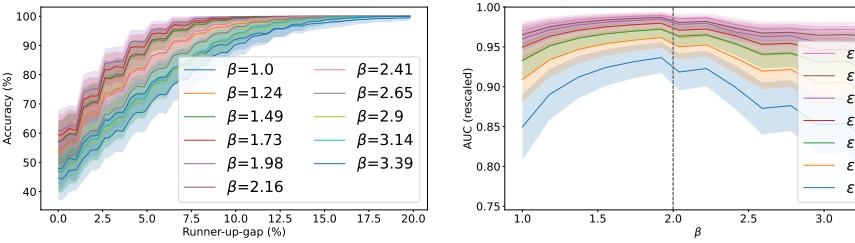

Figure 9: Hardmax Utility for 25-class histogram. Left: Hardmax Utility as a function of runner-up-gap, for mechanisms with equivalent $(2, 10^{-5})$-DP. Right: Area-Under-the-Curve (AUC) of the curves on the left for different values of $\epsilon$, where a higher AUC means a better overall accuracy. AUC is rescaled such that 1.0 is the maximum to normalize across different sizes of domain. We observe that as $\beta$ grows, the more jagged the AUC is, this is because as $\beta$ grows the value of $\epsilon$ becomes more sensitive as a function of $\sigma$ (see Figure 2 for intuition as to why). This observation, coupled with the fact that we compute our Argmaxexperiments for evenly spaced choices of $\sigma$ (and $\beta$) means that more argmax accuracy curves will be grouped together as $\beta$ grows.

We supplement these empirical simulations with a theoretical analysis for the same setting in Appendix D.4, we provide a theoretical analysis of the optimal choice between the Laplace and Gaussian Mechanisms when using a RDP accountant. However, the RDP accountant is known to be suboptimal relative to the PRV accountant used in our experiments.

### D.4 OPTIMAL MECHANISM FOR THE PRIVATE ARGMAX WITH AN RDP ACCOUNTANT

Given oracle access to $\Delta := x_0 - x_1$, the Hardmax-Utility is $\Pr[(x_0 + n_0) > (x_1 + n_1)] = \Pr[(n_0 - n_1) > \Delta]$. This is analytically derivable, as the probability of the sum of two independently drawn random variables is

$$\Pr[(x_0 + n_0) > (x_1 + n_1)] = \Pr[(n_0 - n_1) > \Delta]$$

$$\text{The PDF of } \Pr[(n_0 - n_1) = \int_{-\infty}^{\infty} f_X(x) f_X(\Delta - x) dx$$

$$= k^2 \int_{-\infty}^{\infty} \exp(-\frac{1}{\sigma}(||x||^\beta + ||\Delta - x||^\beta)) dx.$$

Where $k$ is the normalization coefficient for the GG$(\beta, \sigma)$.

This is possible to compute analytically for the Laplace and for Gaussian. For Gaussian, the sum of 2 IID Gaussians, with the same $\sigma$ and $\mu = 0$, is $\mathcal{N}(0, 2 \cdot \sigma^2)$. This means that the Hardmax Utilty for the Gaussian mechanism is $CDF(\mathcal{N}(0, 2\sigma^2)$ evaluated at $\Delta$.

For Laplace, equation 2.3.23 from (Kotz et al., 2001) states that the sum of Laplace random variables with equal $\lambda$ values (centered at $\mu = 0$) is $f_{X_1 + X_2}(x) = \frac{1}{4}\lambda(1 + \lambda|x|) \cdot e^{-\lambda|x|}$. This means that the Hardmax Utilty for the Laplace mechanism is $1 - \frac{1}{4}e^{\lambda \cdot x}(\lambda x + 2)$, evaluated at $x = \Delta$.

The RDP of the Laplace mechanism is $\frac{1}{\alpha - 1} \log[\frac{\alpha}{2\alpha - 1} exp(\frac{\alpha - 1}{\lambda}) + \frac{\alpha - 1}{2\alpha - 1} \exp(\frac{-\alpha}{\lambda})]$. And the RDP of the Gaussian mechanism is $\frac{\alpha}{2\sigma^2}$. This means that for the same $(\alpha, \epsilon)$-RDP guarantee, it is possible to

direclty compute the $\lambda$ and $\sigma$ that let the Gaussian and Laplace mechanism satisfy that RDP guarantee. Given an analytic derivation for the hardmax utility of the mechanism, we can derive when one mechanism is better than another.

This solution gets us to the surprising conclusion that when using the RDP accountant, there are regimes when Laplace is better than Gaussian, regimes when Gaussian is better than Laplace, and regimes where it is data-dependent (depends on the value of $\Delta$).

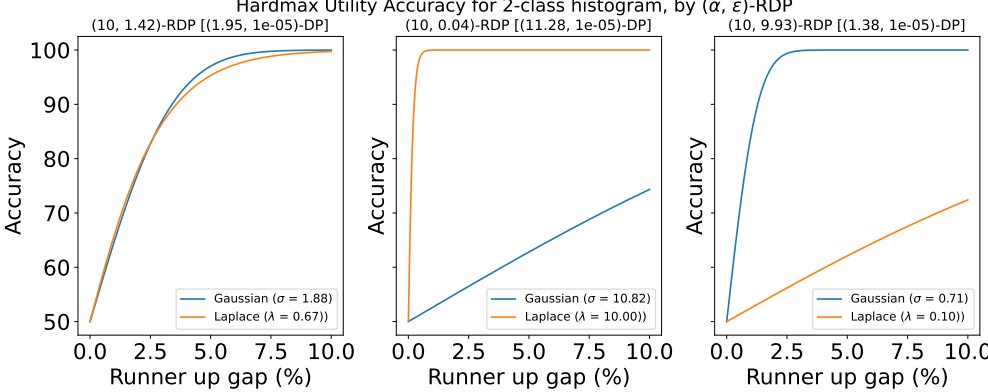

Figure 10: Hardmax Utility for Laplace and Gaussian mechanisms that satisfy as fixed RDP guarantee.

We also do not view the findings under RDP as the result of being a fragile result, but rather as a peculiarity of RDP. The choice of $\alpha$ in RDP can be seen as a parameter for how strongly the tails of a PDF are penalized, where a larger choice of $\alpha$ incurs a larger privacy cost for weight further from the origin. By manipulating the choice of $\alpha$, we can manipulate the relative privacy cost of Laplace Mechanism versus a Gaussian mechanism. As such, it is not surprising that it is possible to choose an $(\alpha, \epsilon)$-pair which favors equivalently-private Gaussian mechanisms, one which favors equivalently-private Laplace mechanisms, or one where they are very similar.

Lastly, in interpretting these findings about RDP, we see no evidence that these RDP findings extend to the PRV accountant, which has been empirically outperforms the RDP accountant and is used as the State-Of-The-Art for privacy accounting. And we do not see these results as a feature of the mechanisms, but rather a peculiarity of a suboptimal accountant with a suboptimal selection of parameters.

# E  ADDITIONAL $\beta$-DP-SGD RESULTS AND IMPLEMENTATION DETAILS

## E.1  $\beta$-DP-SGD ALGORITHM

We now present the $\beta$-DP-SGD algorithm in full form.

---

**Algorithm 6** $\beta$-Generalized Gaussian Differentially Private SGD, $\beta$-DP-SGD($\beta, \sigma, D, l, \eta, L, C, T$)

---

1: **Input:** noise parameters $\beta \geq 1$, $\sigma > 0$, database $D = \{x_1, \ldots, x_N\}$ of points in $\mathbb{R}^d$, loss function $l(\theta, x_i)$, learning rate $\eta$, average group size $L$, clip norm $C$, and training epoch length $T$.
2: Initialize $\theta_0 \in \mathbb{R}^d$ randomly
3: **for** $t = 1$ to $T$ **do**
4:     Construct $L_t \subseteq D$ such that each $x_i \in D$ is included with probability $q = L/|D|$ (Poisson sampling)
5:     **for** each $i \in L_t$ **do**
6:         Compute $G_t(x_i) = \nabla_{\theta_t} l(\theta_t, x_i)$
7:         $\bar{G}_t(x_i) \leftarrow G_t(x_i)/\max\left(1, \frac{\|G_t(x_i)\|_\beta}{C}\right)$
8:     **end for**
9:     Sample $Y_1, \ldots, Y_d \sim_{i.i.d.} \mathcal{N}_\beta(0, \sigma \cdot C)$
10:    $\tilde{G}_t \leftarrow \frac{1}{L}\left(\sum_i \bar{G}_t(x_i) + \vec{Y}\right)$
11:    $\theta_{t+1} \leftarrow \theta_t - \eta \tilde{G}_t$
12: **end for**
13: **Output:** $\theta_T$

---

## E.2    Hyperparameters used in training

**Hyperparameters** We run our $\beta$-DP-SGD algorithm for a maximum of 100 epochs for each parameter setting and sweep over the following parameters: $\beta$ (12 evenly spaced values of $\beta \in [1, 4]$), noise multiplier (6 evenly spaced values of $\sigma \in [0.5, 3.0]$), average batch size $L \in \{128, 256\}$, learning rate $\eta \in \{0.5, 1.0\}$, and clipping norm $C \in \{0.05, 0.1, 0.25, 0.5\}$, for $\delta = 10^{-6}$. Each experiment is run 3 times, which we found sufficient given standard deviations that generally fell below 0.3%.

**Datasets:** We train on CIFAR-10 (Krizhevsky, 2009) and Street View House Numbers (SVHN) (Netzer et al., 2011), two common computer vision datasets, which respectively contain 60,000 and 99,289 small, color images split across ten classes; the Adult dataset (Becker & Kohavi, 1996), a tabular dataset with a binary classification task; and the IMDB dataset (Maas et al., 2011), a collection of movie reviews meant for binary sentiment classification.

**Models:** For the vision classification tasks (CIFAR-10 and SVHN), we use the models described in in Tramèr & Boneh (2020), which previously achieved SOTA results for the $\epsilon \leq\sim 2.5$ regime. Specifically, we train Convolutional Neural Networks (CNNs) described in the original work, and "handcrafted CNNs", which train a CNN on pretrained image features produced by scattering networks (Oyallon & Mallat, 2015). For the the Adult Dataset we train a 2-layer Fully Connected Network (FCN), with 32 neurons in the hidden layer. For the IMDB dataset, we train a Long-Short Term Memory (LSTM) network with 1,081,002 parameters, in order to demonstrate the method on a relatively medium-sized model from scratch.

While the core effect seen in the paper is that there is a very weak relationship with the final test accuracy, it is important to observe that for any one set of hyperparemeters, one particular $\beta$ may perform much better than others.

## E.3    The Role of Individual Hyperparameters in $\beta$-DP-SGD

Below we investigate the role of individual hyperparameters on the the final test accuracy. In general, we observe that while some hyperparameters may have some effect, the general relationship with $\beta$ is unperturbed. Specifically, for many of these experiments the optimal choice of $\beta$ regularly is not $\beta = 2$ (Gaussian), however, it does not tend to deviate far for $\beta = 2$. As in the original training, we emphasize that for some situations plots do not have values for specific choices of $\beta$, this is because the training curves start at a value of $\epsilon$ greater than the $\epsilon$ value we are plotting results for (larger values of $\beta$ tend to consume more privacy per-step, for the same $\sigma$). All of these plots are for $\delta = 10^{-6}$.

### E.3.1 LEARNING RATE

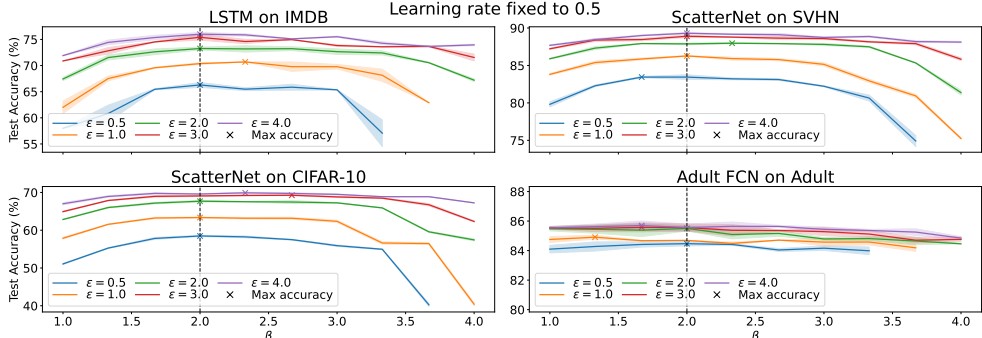

Figure 11: $\beta$-DP-SGD results with fixed learning rate (0.5)

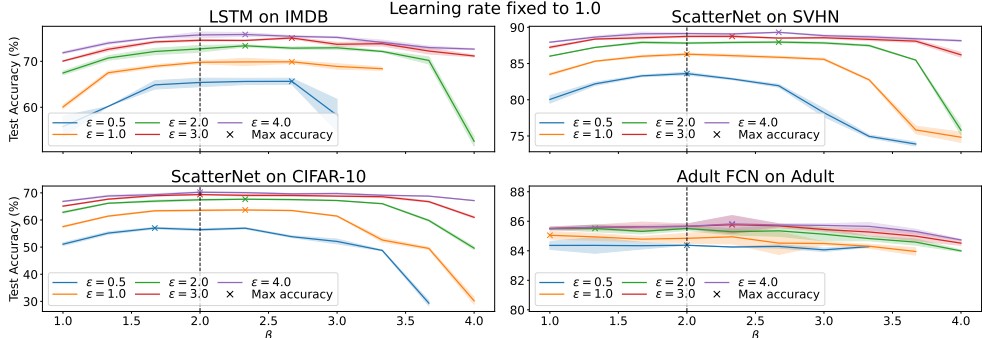

Figure 12: $\beta$-DP-SGD results with fixed learning rate (1.)

### E.3.2 CLIPPING NORM

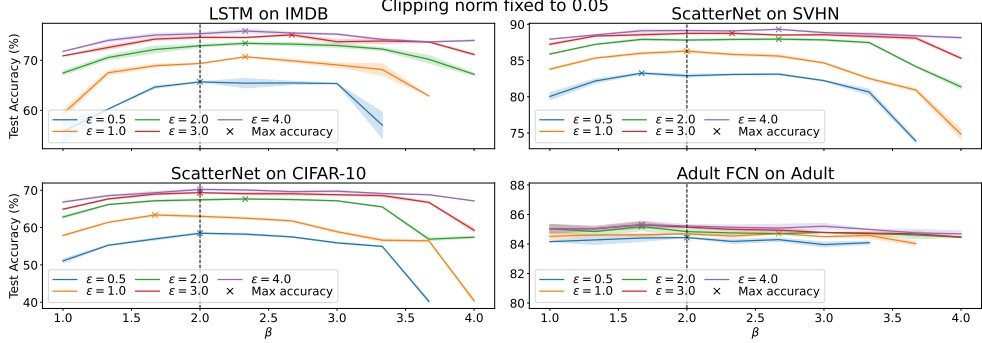

Figure 13: $\beta$-DP-SGD results with fixed clipping norm (0.05)

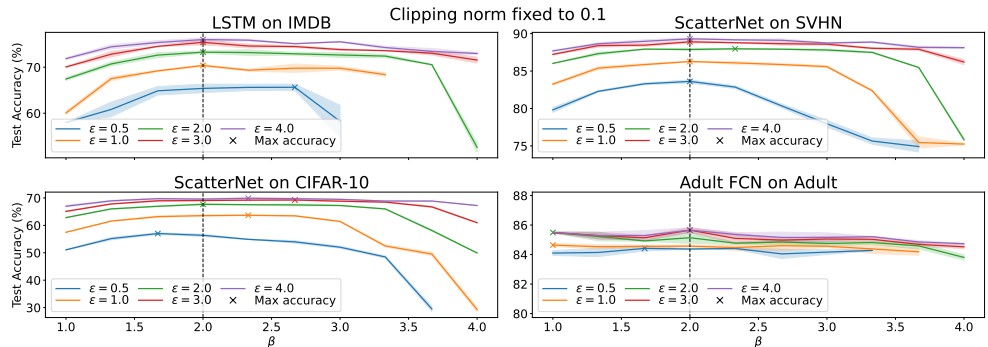

Figure 14: $\beta$-DP-SGD results with fixed clipping norm $(0.1)$

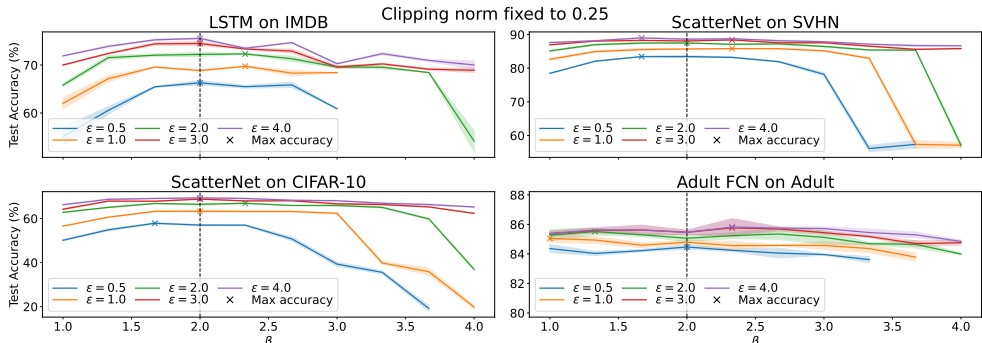

Figure 15: $\beta$-DP-SGD results with fixed clipping norm $(0.25)$

### E.3.3 BATCH SIZE

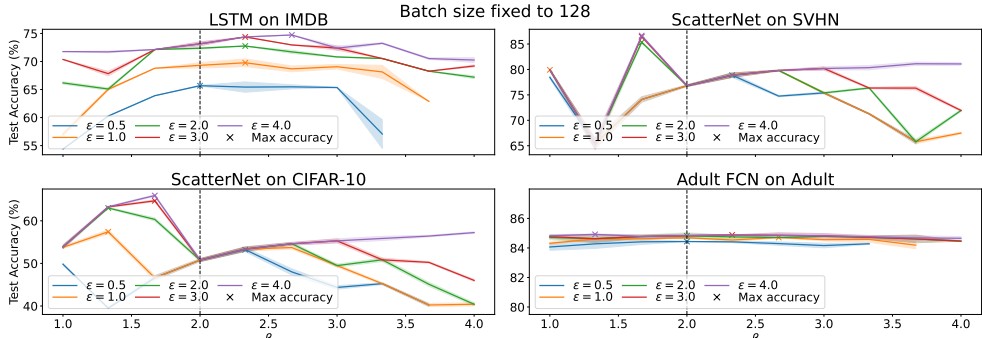

Figure 16: $\beta$-DP-SGD results with fixed batch size $(128)$

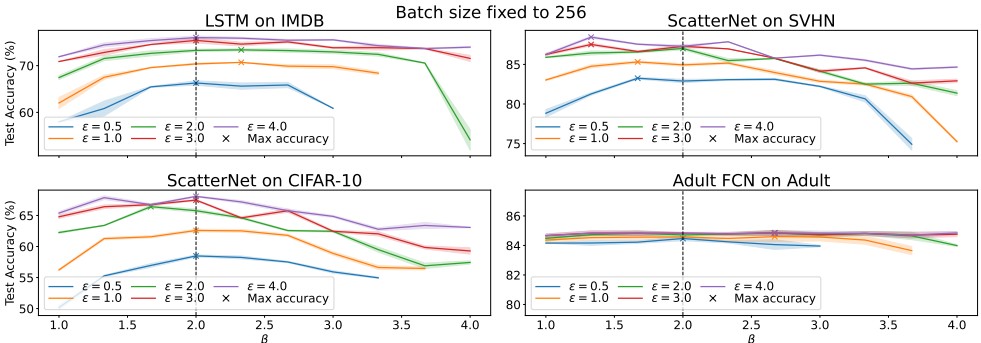

Figure 17: $\beta$-DP-SGD results with fixed batch size (256)

### E.3.4 NOISE MULTIPLIER

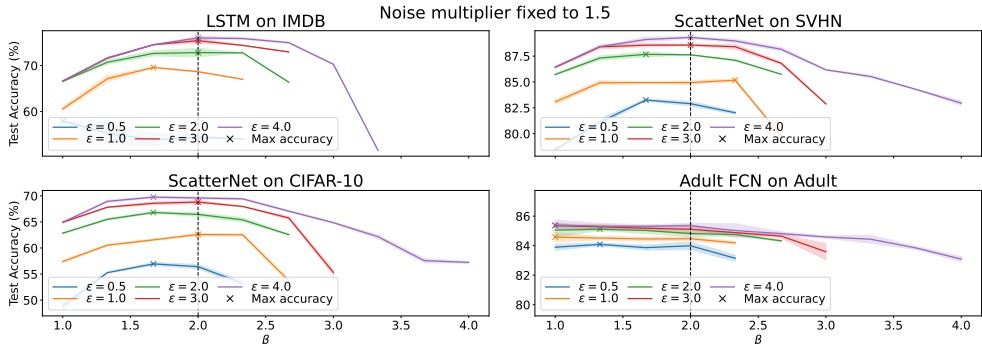

Figure 18: $\beta$-DP-SGD results with fixed $\sigma = 1.5$ (noise multiplier)

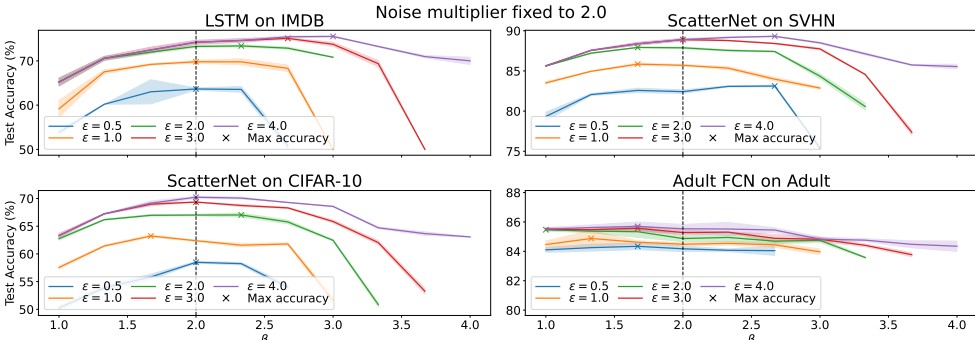

Figure 19: $\beta$-DP-SGD results with fixed $\sigma = 2.0$ (noise multiplier)

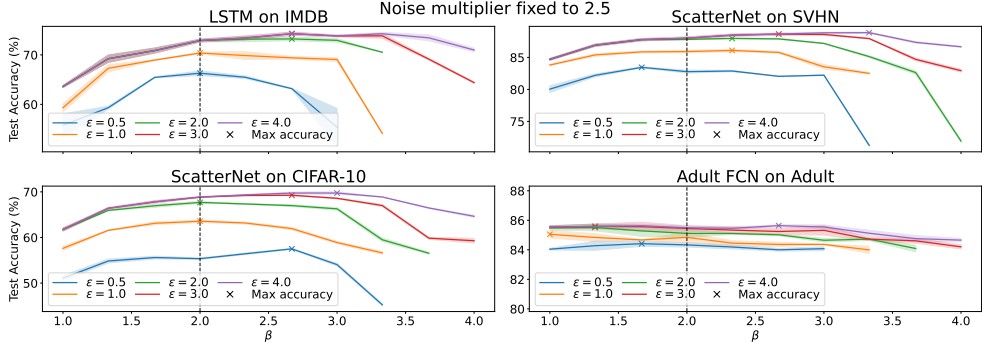

Figure 20: $\beta$-DP-SGD results with fixed $\sigma = 2.5$ (noise multiplier)

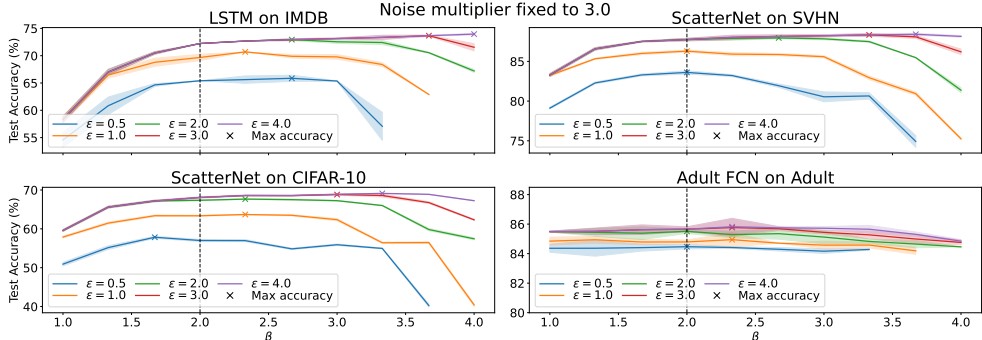

Figure 21: $\beta$-DP-SGD results with fixed $\sigma = 3.0$ (noise multiplier)

### E.4 SAMPLING FROM THE GENERALIZED GAUSSIAN DISTRIBUTION

Unlike the Gaussian and Laplace distributions, sampling from the Generalized Gaussian mechanism is not natively supported by built-in libraries like Python's 'math' library, or the commonly used numpy library. Another commonly used library for statistical computing, SciPy, does have the 'scipy.stats.gennorm' function; however, we found that it regularly takes too long for intensive computations like stochastic gradient descent in practical settings, which involves sampling from high-dimensional gradients thousands of times. Further, the Scipy function is only able to be sampled on a CPU, which makes it ill-suited for DP-SGD, which is regularly performed on a GPU.

We implement a method for sampling from the Generalized Gaussian mechanism included in our code here: `https://anonymous.4open.science/r/GG_for_ml_neurips_code-BD48/README.md`.

In our experiments, we can sample from the Generalized Gaussian only $\sim$ 1.3x slower than sampling from a Gaussian directly. It is possible to conduct similar sampling using the method of inverse probability transforms, since the Generalized Gaussian has a known CDF.

## F REPRODUCIBILITY

### F.1 COMPUTING RESOURCES

For our DP-SGD experiments, the execution of our techniques does not result in a significant increase in processing time compared to the conventional application of DP-SGD. The only addition to the computation duration comes from increased amounts of hyperparameter searching. All experiments and data analysis are reproducible in the codebase provided `https://anonymous.4open.science/r/GG_for_ml_neurips_code-BD48/README.md`.The DP-SGD results in this paper were completed in under 500 hours of GPU time, which was split across 16 machines that were mounted on Nvidia T4 and RTX6000 machines GPUs. All data analysis was conducted on a 8 core machine with 16 GB RAM machine.

