# OpenReview forum: "Beyond Laplace and Gaussian: Exploring the Generalized Gaussian Mechanism for Private Machine Learning"
_ICLR.cc/2024/Conference — Submitted to ICLR 2024_

### Official Review · Reviewer_qfzX · 2023-10-29

**Soundness:** 1 poor
**Presentation:** 4 excellent
**Contribution:** 1 poor
**Rating:** 1
**Confidence:** 5

**Summary:**

This paper investigates the Generalized Gaussian mechanism in Differential Privacy, as extention of typical Laplace and Gaussian noise additions. It proves that the GG family satisfies DP and adapts an existing privacy accounting tool, the PRV accountant, for this purpose. Applying the GG mechanism to PATE and DP-SGD machine learning tools, the authors claim that the Gaussian distribution often optimally balances privacy and accuracy.

**Strengths:**

The flow of this paper is very easy to follow.

**Weaknesses:**

The exploration of alternative distributions for DP in this paper does not offer new insights. Previous research, such as the work by Awan and Dong (2022), already provides a comprehensive study on log-concave and multivariate canonical noise distributions in DP. These studies not only predate but also surpass the current paper's approach by offering tightly derived privacy profiles instead of relying on the numerical experiment method adopted here.

The paper's "Privacy Accounting for GG Mechanisms" relies heavily on numerical experiments without robust error control or rigorous proofs. This methodology is particularly concerning since the privacy profile of this noise family is already well-established in the literature. The paper's focus on a single privacy budget (\(\delta=10^{-5}\)) is both arbitrary and unconvincing. For instance, releasing a \(10^{-5}\) portion of the total data satisfies \(\delta=10^{-5}\) with \(\epsilon=0\), making it theoretically superior to all other mechanisms discussed in terms of this metric.

The experiments (PATE, DP-SGD) conducted under the privacy budget of \(\delta=10^{-5}\) are too limited to convincingly demonstrate the superiority of any specific \(\beta\) values. At this \(\delta\) setting, the noise variance added by the mechanisms varies widely, which is very likely to be the true reason behind the differing outcomes.

For example. for a fixed \(\epsilon\), the Gaussian mechanism will perform poorly for very small \(\delta\) values as the required variancediverge rapidly. In contrast, for the Laplace mechanism, the scale of the Laplace noise remains approximately unchanged. Conversely, when considering Gaussian Differential Privacy (GDP) as the privacy budget, the Gaussian mechanism generally outperforms the Laplace mechanism. There is no intrinsic advantage or disadvantage for either of these two algorithms; their efficacy largely depends on the specific form of the privacy constraint. The statement "This provides a justification for the widespread adoption of the Gaussian mechanism in DP learning" seems coincidental within the scope of this research approach. The preference for the Gaussian mechanism may be more attributed to its strong alignment with GDP accounting, rather than any inherent superiority deduced from the methods used in this research.

It is incorrect (vastly loose) to compute composition privacy budget from a single pair of epsilon and delta (think about composition of Gaussian mechanisms).


In Section B.2, titled "Mechanisms with Equivalent Privacy Guarantees," the selection of \(\sigma\) is derived through random search, not through analytical computation. This approach does not guarantee the avoidance of numerical stability issues (which is very likely to happen for very small $\delta$ when using the PRV accountant's random search to determine \(\epsilon\). The method should at least be executed using well-established mechanisms (Gaussian and Laplace) to validate the outputs against their analytically true values.

Some citations in this paper can be improved to be more relevant. For example, (Kairouz, P., Oh, S., & Viswanath, P. (2015)) as optimal compostion for epsilon,delta DP is more suitable than (Abadi et al., 2016) which is not directly related to the topic after definition 4.

There are some confusions in the proof and presentation (see questions).
For example:
Holder’s Inequality is listed as a lemma
Page 19: "As x goes to infinity, the power of the exponent is negative". It might be better to say " ... is negative for sufficiently large x in terms of ..."
Also see questions




There is some broken link (for example, at the beginning of C.6).

Kairouz, P., Oh, S., & Viswanath, P. (2015). The composition theorem for differential privacy. In International conference on machine learning (pp. 1376-1385). PMLR.

Awan, J., & Dong, J. (2022). Log-concave and multivariate canonical noise distributions for differential privacy. Advances in Neural Information Processing Systems, 35, 34229-34240.

**Questions:**

What is the purpose for Figure 11? What are the pattens to be shown here?

On page 23 what does "bound the probability of not being (ϵ, 0)-DP" mean? Being DP or not is an intrinsic property of an algorithm, where does the randomness come from?

What is a "'single composition' of the GG mechanism"?

---

> ### Author Response · Authors · 2023-11-19
>
> We thank the reviewer for their time and respond to points below (in 2 comments).
>
> Thank you for pointing the paper from Awan and Dong (2022) to us, we now incorporate it into the updated paper. Generally we do not believe there are any conflicts between the works; instead both complement each other and point to the same direction –  DP could gain a boost from a correctly tuned noise distribution. More in-detail analysis of the differences is given below:
> * The work by Awan and Dong is both important and very impressive, but our works differ in some key ways. Their work is an important work exploring the theoretical foundations of privacy mechanisms, but does not actually answer the question of what mechanisms are optimal for private machine learning.
> * While Canonical Noise Distributions (CNDs) are a great notion, they are fundamentally a privacy guarantee, and do not make a claim about the utility of a mechanism. Our work looks at the privacy-accuracy tradeoff in machine learning for a family of mechanisms - something otherwise minimally studied. While intuitively it may seem that the CND should be an optimal mechanism because they minimize noise added for a given f-DP trade-off function, this assumes that all parts of the tradeoff functions are equally valuable to the utility. Imagine a utility function that strongly punishes outliers - in that setting, you want a mechanism that minimizes noise above some level, rather than more than one that minimizes total noise. We explore this in Appendix B.5.
> * Proposition 4.2 and Theorem 4.3 only provide CNDs for l_infinity and l_1 bounds (respectively). L_infinity is an uncommon sensitivity, and not used in practice, while our own experiments using multivariate Laplace (GG for beta =1) lead us to believe that l_1 sensitivity with IID noise is suboptimal from an accuracy perspective.
> * Privacy accounting is critical for tracking privacy during private ML. Their paper does provide theorems for composition of the CND mechanisms in f-DP, however, it remains unclear how to tightly compute the privacy of CNDs to compute an epsilon-delta bound using a known privacy accountant. Our work provides an accountant for each member of the GG family.
> * Subsampling has been critical for making private ML applications work for reasonable privacy budgets. Their work does not directly consider how to compute the proper CND for the subsampled setting, nor how to do privacy accounting for subsampled variants of mechanisms based on CNDs.
> * The two most useful DP formulations are GDP and  F_{epsilon,delta}-DP. However, the paper does not build a CND for F_{epsilon,delta}-DP (Section 4.4) which is usable for ML. Their mechanism does not consider subsampling, and only considers the CND when using l_infinity sensitivity.
> * They show that for Gaussian-DP, Gaussian mechanisms are CNDs. However, the GDP Accountant can significantly underreport the true \epsilon (https://arxiv.org/abs/2106.02848), and so unfortunately this does not directly translate into a useful application.
>
> **Below we address other specific comments in the review.**
>
> `` The paper's "Privacy Accounting for GG Mechanisms" relies heavily on numerical experiments without robust error control or rigorous proofs.``
> * We address the error of the privacy accountant in Appendix B.3. More specifically, Theorem 6 provides an error analysis of the privacy accountant, which provides an arbitrarily tight privacy accountant. (We set our acceptable tolerance for \epsilon to be 0.01, and round our reported values up to the nearest 0.01.)
>
> ``The paper's focus on a single privacy budget ((\delta=10^{-5})) is both arbitrary and unconvincing. For instance, releasing a (10^{-5}) portion of the total data satisfies (\delta=10^{-5}) with (\epsilon=0), making it theoretically superior to all other mechanisms discussed in terms of this metric…. The experiments (PATE, DP-SGD) conducted under the privacy budget of (\delta=10^{-5}) are too limited to convincingly demonstrate the superiority of any specific (\beta) values. At this (\delta) setting, the noise variance added by the mechanisms varies widely, which is very likely to be the true reason behind the differing outcomes. ``
>
> * We thank you for this comment, we actually run our DPSGD experiments with delta=1e-6. However, it appears that in our paper, we wrote 1e-5 in one place (Appendix E.2), as a vestige of a previous version. We have updated this, thank you for catching this.
> * It’s worth also noting that presenting results for a single choice of delta= 1e-5 is standard for many DPSGD experiments. Two well-cited and well-regarded papers that achieve SOTA results use delta=1e-5 for CIFAR-10  https://arxiv.org/abs/2011.11660  and ​​https://arxiv.org/abs/2204.13650.
> * Regarding the comment that at this delta, the variance varies widely. We are unsure what the reviewer is referring to, as the PRV accountant is correct to arbitrary precision.
>
> [1/2]

---

> > ### Author Response · Authors · 2023-11-19
> >
> > ``There is no intrinsic advantage or disadvantage for either of these two algorithms; their efficacy largely depends on the specific form of the privacy constraint. The statement "This provides a justification for the widespread adoption of the Gaussian mechanism in DP learning" seems coincidental within the scope of this research approach.’’
> > * In Machine Learning applications of DP, there is an explicit advantage to choosing, e.g., Laplace over Gaussian noise because in practice they have different privacy-utility functions. PATE (https://openreview.net/pdf?id=rkZB1XbRZ) for example, explicitly cites the differences in privacy-accuracy tradeoffs and specifically the likeliness-of-outliers (for the same RDP accountant) as the reason for switching from laplace to gaussian noise.
> > * For a particular delta and composition count, one incurs a particular tradeoff in weight-in-tails vs weight-in-body; our findings show that surprisingly, this does not contribute to a noticeable effect in DPSGD. See Appendix B.5 for one concrete measure of how this tradeoff occurs.
> >
> > ``The preference for the Gaussian mechanism may be more attributed to its strong alignment with GDP accounting, rather than any inherent superiority deduced from the methods used in this research. ``
> > * The PRV accountant is not designed with a particular mechanism in mind, and as far as we know there is no connection between GDP and PRV accounting.
> > * Further, in the PRV paper (https://arxiv.org/pdf/2106.02848.pdf) they show that GDP accounting regularly underestimates privacy (as opposed to the PRV accountant).
> >
> > ``It is incorrect (vastly loose) to compute composition privacy budget from a single pair of epsilon and delta (think about composition of Gaussian mechanisms).``
> > * We agree. Is there a particular location where the reviewer sees this to be a problem? If the reviewer is referring to the fact that we produce a privacy curve for the privacy consumed in Figure 1, then we emphasize that this plot is there for illustrative purposes and that this plot would be qualitatively the same for a different choice of composition of (epsilon, delta) pairs (and not just a single pair).
> >
> > ``In Section B.2, titled "Mechanisms with Equivalent Privacy Guarantees," the selection of (\sigma) is derived through random search, not through analytical computation.``
> > * There is a misunderstanding here: we propose selecting sigma through a binary search rather than a random search. Binary search is a reasonable approach here because for fixed \beta and \delta, increasing \sigma increases \epsilon, making the privacy-sigma curve monotonic. Again we emphasize that the privacy accountant computes epsilon with arbitrary precision (shown in B.3).
> > Further, in practice we set (beta, sigma, sample rate q, compositions N, delta) and then compute the privacy accounting for that, rather than fixing epsilon, and solving for sigma.
> >
> > ``What is the purpose for Figure 11? What are the patterns to be shown here?``
> > * We felt that it helps see how the pareto-optimality curve emerges, and one set of hyperparameters for optimal results at epsilon = 1 does not necessarily correspond to the hyperparameters for optimal test accuracy at epsilon = 2.
> >
> > ``On page 23 what does "bound the probability of not being (ϵ, 0)-DP" mean? Being DP or not is an intrinsic property of an algorithm, where does the randomness come from?``
> > * Thank you for catching this. We should have been more precise in our language, and have updated this text accordingly. Informally, we meant to say that we bound the probability of choosing a value from the support that would not satisfy `P(M(x) =y) < e^eps P(M(x’)=y)`. It is described in these notes in Lemma 1.4 https://dpcourse.github.io/2021-spring/lecnotes-web/lec-09-gaussian.pdf
> >
> > ``What is a "'single composition' of the GG mechanism"?``
> > *By this, we mean that the GG mechanism is applied once (N=1 compositions)
> >
> > Regarding the concrete points on citations and language, we thank you for these comments and introduce them now.
> >
> >
> > Thank you again for your time and comments.
> >
> > [2/2]

---

### Official Review · Reviewer_QnfG · 2023-10-30

**Soundness:** 2 fair
**Presentation:** 3 good
**Contribution:** 1 poor
**Rating:** 3
**Confidence:** 3

**Summary:**

This paper studies the generalized Gaussian mechanism for private machine learning, especially DP-SGD and PATE. The generalized Gaussian mechanism is a mechanism utilizing the generalized Gaussian noise that encompasses Laplace, Gaussian, and arbitrary noise distribution associated with the density function proportional to $e^{-\frac{|x-\mu|^\beta}{\sigma}}$. The paper investigates the optimal $\beta$ in the generalized Gaussian in terms of utility-privacy trade-off in DP-SGD and PATE. To this end, the authors proposed $\beta$-DP-SGD and GGNMax algorithms which are variants of DP-SGD and LNMax by replacing the noise distribution with the generalized Gaussian. The authors show empirically that choosing $\beta=2$ is near-optimal in test accuracy.

**Strengths:**

This paper is well-written and well-organized. I found that investigating the optimal parameter $\beta$ of the generalized Gaussian mechanism for private ML is interesting and important in ML society. I found that $\beta=2$ is near-optimal is very interesting, and I think it opens other research directions.

**Weaknesses:**

Generally, I do not see much contribution in this paper. Several things I am concerned about are listed:
- The proposed mechanisms ($\beta$-DP-SGD and GGNMax) are a straightforward generalization of the existing mechanisms by replacing noise distribution.
- The analysis using the PRV accountant is not novel.
- I felt that there are not many analytical results in the generalized Gaussian mechanisms. Most of the results are empirical findings, and I doubt that the experiments are sufficient to claim the findings.

**Questions:**

Main questions:
1) Is Theorem 1 saying there exists $\epsilon$ and $\delta$ for $(\epsilon,\delta)$-DP? Isn't it obvious that every mechanism has $\epsilon=1$ and $\delta=1$ if I am not mistaken?
2) For "However, unlike the results ... larger than $\beta>3$" on page 8, is there any reason why this happens?
3) If I am not mistaken, the comparisons regarding $\beta$ are done after hyper-parameter optimization. If they are, I have some questions:
- Is it reasonable to perform the comparison by fixing hyperparameters that are optimal in non-DP training?
- For fixed hyperparameters, $\beta=2$ is still near-optimal? or is it dependent on hyperparameters?

Some minor questions are as follows:
1) In the caption of Figure 4, it is written that the reported three epsilons are based on the minimum epsilon giving $0.98$ accuracy. Why then the accuracy is much worse than $0.98$ in the plots even for $\epsilon^\prime$?
2) Renyi DP is defined in the main paper but is not used as a main part after that. Is there any reason for defining it in the main paper?

---

> ### Author Response · Authors · 2023-11-19
>
> We thank the reviewer for their time and comments. We respond to questions and weaknesses, separately below, starting with the questions. (We respond in 2 parts)
>
> ``Is Theorem 1 saying there exists eps  and delta  for (eps,delta)-DP? Isn't it obvious that every mechanism has eps=1 and delta = 1  if I am not mistaken? ``
> * You are correct that if delta is allowed to be 1, then all mechanisms are (epsilon, delta)-DP; however Theorem 1 states that there is an epsilon value for any choice of beta, sigma, and delta. While this seems intuitive in some cases it is certainly not true for all algorithms, and this precondition is a prerequisite to using the PRV Accountant. As an example, a mechanism that runs the \epsilon-DP Laplace mechanism with probability 0.9, and with the remaining 0.1 probability returns the non-noisy answer is (\epsilon, \delta)-DP for some \delta, but not for any delta < 0.1.
>
> ``"However, unlike the results…. larger than beta> 3 " on page 8, is there any reason why this happens?``
> * First, we emphasize that the label accuracy decline is on the order of ~0.1%, and so this is not a significant drop. The drop is most noticeable specifically because of how little variance there is in the label accuracy across values of \beta. That said, we believe the drop is due to an artifact of how the mechanisms of equivalent DP guarantees are generated. To describe it in detail: first we add noise from an evenly distributed grid of \beta \in [1,4] and \sigma \in [0.01,7] values, and then we compute the privacy levels for those (\beta,\sigma) tuples and compute the corresponding mechanisms of approximately equal DP guarantees. Because of this, we overestimate \epsilon when solving for mechanisms with equivalent DP guarantees; this is most noticeable for larger choices of \beta, where the same difference in \sigma contributes to a larger \epsilon.
>
> ``Is it reasonable to perform the comparison by fixing hyperparameters that are optimal in non-DP training? ``
> * Previous work has shown that hyperparameters which are optimal in non-DP training are not necessarily optimal for DP training. This was explored in this work https://arxiv.org/abs/2007.14191, and a later work https://arxiv.org/abs/2110.03620 produced a scheme for how to do privacy preserving hyperparameter tuning.
>
> ``For fixed hyperparameters, beta=2  is still near-optimal? or is it dependent on hyperparameters?``
> * In general, it is near optimal, but it does depend on the hyperparameters, we provide plots for this in Appendix E.3.
>
> [1/2]

---

> > ### Author Response · Authors · 2023-11-19
> >
> > **We respond to the weaknesses**
> >
> > ``The proposed mechanisms (beta-DP-SGD and GGNMax) are a straightforward generalization of the existing mechanisms by replacing noise distribution.``
> > * While we agree that there are many new mechanisms to explore, to the best of our knowledge, our work is the first to study privacy-accuracy tradeoffs with alternative noise distributions for DPSGD or PATE. The GG family is a natural place to start, because it contains both Laplace and Laussian mechanisms as special cases, but should be seen as the start of a line of work, not the end.
> > * We view the relation to existing mechanisms as a strength rather than a weakness. Our results show that even minor changes to existing commonly-used algorithmic structures can provide empirical improvements, and that existing private ML algorithms can be easily extended to accommodate optimally-tuned noise from the GG mechanism.
> > *We emphasize that our main contributions are not novel algorithmic developments, but rather the analysis and application of the GG family of mechanisms to ML applications of interest through these existing algorithms (DP-SGD and PATE).
> >
> > ```The analysis using the PRV accountant is not novel.```
> > * The PRV accountant is impressive work that provides a huge step forward in privacy accounting. However, prior to our work the PRV accountant could only be used with mechanisms with known privacy loss random variables, and their CDFs -- something that is generally non-trivial to compute. While it is intuitive to numerically compute this, we provide arbitrarily tight error bounds, showing that this is acceptable to do, and we provide the sample complexity of doing so. Further, in our new version of the paper, we provide a proof that the GG mechanism (with l_beta sensitivity) is dimension independent (in Appendix B.2), allowing for much easier accounting. To the best of our knowledge, both of these ideas are novel and not previously known.
> >
> > ``I felt that there are not many analytical results in the generalized Gaussian mechanisms. Most of the results are empirical findings, and I doubt that the experiments are sufficient to claim the findings.``
> > * We agree that more theoretical guarantees would be nice but for both GGNmax and DPSGD this is problematic for two reasons. 1. The best privacy accountants are numerical, and thus prevent us from making a statement about a privacy-accuracy tradeoff. 2. For DPSGD it is exceedingly difficult to make theoretical statements about non-convex problems. These two reasons make it very difficult to make a theoretical statement. However, for the GGNmax setting, we attempt to get around this by providing numerical experiments from PATE (main body), simulations from a larger family of histograms (Appendix D.3), and theoretical analysis when using the RDP accountant (Appendix D.4).
> >
> > Minor questions:
> > ```In the caption of Figure 4, it is written that the reported three epsilons are based on the minimum epsilon giving 0.98 accuracy. Why then is the accuracy is much worse than 0.98 accuracy  in the plots even for eps’?```
> > * In our submission, we caught that there was a potential improvement by switching from l2 to l_beta sensitivity. As such, we have regenerated Figure 4, and we show our plots for fixed epsilons. Thank you for catching this, and we hope the new changes address your concern.
> >
> > Thank you very much for your time and comments.
> >
> > [2/2]

---

> ### Comment · Reviewer_QnfG · 2023-11-22
>
> Thank the authors for the detailed response to my questions. I have a better understanding of your work now. The motivation and investigating the optimized $\beta$ in the GG mechanism are interesting and meaningful.
> Nevertheless, I still doubt if this paper is an ICLR-quality paper. The main reason is the lack of theoretical or rigorous supporting evidence for the claim ($\beta=2$ is near optimal). I understand that the analysis of the generalized GG mechanism on DP-SGD is quite problematic. However, even the empirical results in the paper are not sufficient in my opinion. For such a reason, I maintain my score.

---

> > ### Author Response · Authors · 2023-11-22
> >
> > Dear reviewer,
> >
> > Thank you for your updated response. Could you possibly clarify what concerns are still remaining?
> >
> > We wish to emphasize the context in which this work comes in. The field of private mechanisms for ML is largely unexplored, and as far as we know, no prior work has provided any insight into privacy-utility tradeoff for alternative DP mechanism for DPSGD. Awan and Dong (2022) (https://arxiv.org/abs/2206.04572) and the Alghamdi et al (2022) (https://arxiv.org/abs/2207.00420) choose utility measures based on theoretical intuition, and derive optimal mechanisms for that; however, their works are purely theoretical and remain difficult to implement - thus there remains a large gap in understanding the privacy-accuracy tradeoff of private ML with alternative mechanisms. See our response to qfzX for a more thorough comparison about Awan and Dong (2022) (https://openreview.net/forum?id=JG9PoF8o07&noteId=hhs9N8UCYe). No prior work has even compared DPSGD with the Gaussian mechanism to a variant of DPSGD with the Laplace mechanism.
> > What experiments or theory would you like to see that you would find convincing?
> >
> > We also wish to call attention to that in our new update we show that the privacy accounting for the GG mechanism is dimension-independent.  We do not believe this is true for other families; meaning that we are able to do privacy accounting for the GG mechanism significantly more simply than any other mechanism. We emphasize that adding noise proportional to dimension, or having to do privacy accounting proportional to dimension, will either add losses in utility or huge computational costs when most neural networks have millions (if not more) parameters. As such, this family of GG mechanisms would perhaps be seen as the most promising direction for alternative mechanisms; however, our results show that within this family, the Gaussian mechanism either performs nearly-as-well or outperforms other members of the GG family.
> >
> > Further, we call attention that our work is not constrained to DPSGD, but extends to PATE  (Private Aggregation of Teacher Ensembles) as well. We again emphasize that there are not clear tools for comparing privacy-accuracy tradeoffs when using numerical accounts; however we provide empirical results for how GGNMax performs on the intermediate outputs of PATE as well simulation results for a larger class of families that show. We supplement this with a rigorous theoretical analysis of how these results change when using RDP, which is known to be less tight than the PRV, in appendix D.4.
> >
> > Are there any experiments or theory that you would like to see that you would find convincing for PATE, DPSGD, or some other part of the work?
> >
> > Once again, thank you for your response.

---

### Official Review · Reviewer_7vy3 · 2023-10-31

**Soundness:** 3 good
**Presentation:** 4 excellent
**Contribution:** 3 good
**Rating:** 8
**Confidence:** 2

**Summary:**

This paper takes a deeper look at the generalized Gaussian mechanism.
The first contribution is to extend the privacy proof of the GGM to work with the PRV accountant to allow tighter analysis than previous bounds.
They then plug the mechanism into various ML applications and observe the effect of varying beta on accuracy.
In general, they find support for using the Gaussian mechanism, but also that more fine-tuned versions of beta (fractional values) can lead to slight improvements in accuracy.

**Strengths:**

- Even though the results didn't yield significant changes while varying beta, I see great value in a study that supports the approximate optimality of the Gaussian mechanism. In general, there are so many hyperparameters to tune, so giving practitioners support in using the Gaussian mechanism as a default is useful.
- Very well-written paper with clear descriptions.

**Weaknesses:**

- Claiming STOA with three runs: The results on machine learning had quite a high variance due to limited runs. I think claiming STOA when the accuracy is better by such a small amount is not significant. Perhaps more runs and a hypothesis test/confidence interval would give more definitive results (although STOA is not the primary goal of the work).

- One more hyperparameter: the very small gains in accuracy would surely be outweighed by the cost of tuning beta in practice. I see the main contribution of this work is showing that the beta doesn't have too much effect rather than improving the accuracy of current mechanisms.

### Typos
- End of Section 3.2, last paragraph "change subsequently change".
- A.1 Proof of theorem 4 has a missing ref.

### Note
I did not verify the proofs in this paper as I do not have sufficient theoretical background. I apologize that I have to leave this task to my fellow reviewers.

**Questions:**

- What would be the solution to fix the artifacts in Figure 3? Perhaps more values in the grid?

---

> ### Author Response · Authors · 2023-11-19
>
> We thank the reviewer for the thorough comments and the kind words. We address the weaknesses stated in the review below:
>
> * Regarding the state-of-the-art results: as you stated, achieving state-of-the-art was not the primary goal of the work, and rather only a baseline to compare against; given this goal, we found that 3 runs were sufficient because the resulting standard deviations in nearly every case was less than 0.3%.
> * We agree that this is one more hyperparameter to tune, and generally agree that “I see the main contribution of this work is showing that the beta doesn't have too much effect rather than improving the accuracy of current mechanisms.” We also agree, as you state, that “there are so many hyperparameters to tune, so giving practitioners support in using the Gaussian mechanism as a default is useful.”
>
> We have addressed the typos, thank you for pointing those out.
>
> Thank you very much for your time and comments.

---

> > ### Comment · Reviewer_7vy3 · 2023-12-04
> > **Thanks for the response**
> >
> > I would like to thank the authors for their response. I agree with their comments and my review was already very positive, so I leave it unchanged.

---

### Official Review · Reviewer_aH3Q · 2023-10-31

**Soundness:** 4 excellent
**Presentation:** 4 excellent
**Contribution:** 2 fair
**Rating:** 5
**Confidence:** 3

**Summary:**

In differential privacy literature, most papers focus on either Laplace or Gaussian Mechanism due to the simplicity of analysis and proven effectiveness in practice. This paper studies the Generalized Gaussian Mechanism instead. It shows that the mechanism is differentially private and through experiments on common benchmarks, it also shows the effectiveness of the mechanism on deep learning tasks.

**Strengths:**

- The paper provides some empirical evidence on why Gaussian Mechanism is popular. In their experiments, the model usually achieves the best accuracy when the noise added is roughly Gaussian.

- The Generalized Gaussian Mechanism can be useful for tasks that require very high accuracy. By tuning the $\beta$ carefully, it's possible that the model can achieve better results (as shown by some improvements on cifar10 tasks).

- The paper is well-written overall.

**Weaknesses:**

- The experiments are conducted on fairly small datasets and deep learning architectures so it's hard to get a grasp of how well the generalized gaussian mechanism works.

- While the study is fairly interesting, I'm not sure if the technical contribution is enough. The main takeaway is Gaussian and Laplace are pretty much the best choices as expected. Generalized Gaussian Mechanism is fairly hard to analyze since it doesn't have a closed-form relationship between $\beta$ and $\epsilon, \delta$. The experiment also requires the tuning of $\beta$ which in the end ends up being Gaussian and Laplace mechanism anyway.

**Questions:**

- In page 18, I think the RHS should be $(\frac{\alpha-1}{\alpha})^{1/\beta}$?

- Also as $x<0$, shouldn't $\frac{|x|}{|x-\mu|}$ be $\frac{-x}{\mu-x}$ instead? Fortunately, I don't think it affects the argument.

---

> ### Author Response · Authors · 2023-11-19
>
> We thank the reviewer for their response. We appreciate your feedback. Below we respond to the weaknesses stated in the review.
>
> ``The experiments are conducted on fairly small datasets and deep learning architectures so it's hard to get a grasp of how well the generalized gaussian mechanism works.``
> *  Unfortunately current research on private machine learning is still developing techniques for accurately training on larger datasets. The error contributed by differential privacy scales with dimension (https://arxiv.org/pdf/2106.00001.pdf), and larger datasets tend to require larger models to perform well. This is evidenced by the fact that the state-of-the-art results for CIFAR-10 are still below 70% accuracy for epsilon =1 (which is often considered an acceptable privacy risk).
>
> ``While the study is fairly interesting, I'm not sure if the technical contribution is enough``
> * Prior to our work, very little research has explored alternative differential privacy techniques for private machine learning. One notable exception is the Cactus mechanism (https://arxiv.org/abs/2207.00420), however, it is not implementable for models with a large number of parameters (>40). As far as we are aware, no work exists even considering Laplace noise for DP-SGD. Yes, our work finds that the Gaussian mechanism is close-to-optimal for the family of privacy mechanisms we explore. However, we emphasize our other contributions:
>     * We provide the first exploration of DP mechanisms outside of the Gaussian Mechanism for DPSGD for private ML and provide a practical implementation of this mechanism for private ML.
>     * We provide a framework for comparing Laplace and Gaussian mechanisms, as well as how to use existing privacy accountants for arbitrary mechanisms.
>     * To us, the authors, it is exceedingly surprising that the Gaussian Mechanism performs close to optimally. It’s hard to apply a historical rationale for why previous researchers only considered the Gaussian mechanism, but it seems likely that this is because they were familiar with it mathematically, and it has nice properties for computing integrals. However, neither of those two reasons are any reason why the Gaussian mechanism (out of the infinitely many other possible mechanisms) should perform optimally. As a field, we have explored very little of the space of potential DP mechanisms, and to find that one of the two original mechanisms were in fact near optimal, is at the very least surprising.
>     * In our exploration we provide a _mostly_ negative result, that researchers should mostly not explore other unimodal DP mechanisms for DPSGD. While positive results are easier to accept, a negative result (that Gaussian noise performs near-optimally) is nonetheless a significant step forward in our understanding of privacy-accuracy tradeoffs in machine learning.
>
> Regarding your questions - yes, we believe you are correct, and we have updated it accordingly. As you noted, we also find that this does not change the argument.
>
> Thank you very much for your time and comments.

---

### Author Response · Authors · 2023-11-19

We thank the reviewers for all their comments. We are submitting an updated version of the paper alongside individualized comments (major text updates are in green). The primary update presented is that we realize that our \beta-DPSGD algorithm uses fixed sensitivity (using the l2 -norm), and that changing it to use `l-\beta` allows for a significant gain in the complexity of doing privacy accounting. It also enables us to compare the Laplace mechanism (typically with l1 norm) and the Gaussian mechanism (typically used with l2 norm) on the same playing field. We provide a proof that \beta-DPSGD with sensitivity in the l-\beta norm is dimension-independent in Appendix B.2. We updated our \beta-DPSGD algorithm, and also the associated experiments for \beta-DPSGD (Figure 4) as well as the results in the appendix. We note that while this does change the specific results, the trends and effects remain effectively completely unchanged. We also modified our manuscript to address comments made by the reviewers.

---

### Meta-Review · Area_Chair_HeoC · 2023-12-11

**Metareview:**

This paper investigates the use of generalized Gaussian (GG) mechanisms in differentially private machine learning. The core motivation is that, by varying the distribution of the noise added to data queries (i.e., changing the norm $\beta$ in the exponential distribution), one has access to an additional hyper-parameter for tuning in order to achieve better accuracy and privacy. Some evidence is provided that norms $\beta$ other than $\beta=1$ (Laplace) and $\beta=2$ (Gaussian) can render gains in both privacy and accuracy in methods such as PATE and DP-SGD.

The paper is clearly motivated, and the authors do a good job in positioning their work within existing literature. I found the appendices particularly informative.

The reviewers noted several limitations of the paper. One axis of concern is the contributions. The paper is largely experimental, finding little evidence to use distributions beyond Gaussian. Though a negative result is welcome, there were also concerns with the experimental benchmarks supporting the paper's claims. It is difficult to see how one can claim Gaussian is near "optimal" from the experiments given in the paper. For example, in Figure 3, error bars are indistinguishable -- it is not evident what statistically significant claim can be taken from this experiment.

Moreover, there were also concerns regarding privacy accounting with the GG mechanism. Both in appendix B.2 and appendix C.4, the authors discuss the dimension independence and the computation of the PLRV of the mechanism. Note that, in both cases, the definition of "neighboring" in differential privacy has to be adapted to match the chosen norm $\ell_\beta$. This fundamentally changes the privacy guarantees offered by each mechanism since the shift changes, and, consequently, the *de facto* guarantees also change. Such issues need to be more precisely addressed in the paper.

One reviewer (qfzX) gave a strong reject score to the paper. My own reading of the paper indicates that this score is overly harsh, and this was taken into account in the decision.

After discussion, the reviewers' concerns regarding the contributions and experiments remained. This sentiment was captured by reviewer QnfG, who stated that the primary issue is the absence of substantial theoretical or empirical evidence supporting the claim of near-optimality. The analysis of the GG mechanism on DP-SGD is insufficient to draw general conclusions, and the empirical results provided do not adequately substantiate the assertions.

**Justification For Why Not Higher Score:**

Overall, the paper need to be further developed -- both in terms of theoretical and practical developments -- prior to publication.

**Justification For Why Not Lower Score:**

N/A

---

### Decision · Program_Chairs · 2024-01-16

Reject